# Enhanced Hyperspectral Sharpening through Improved Relative Spectral Response Characteristic (R-SRC) Estimation for Long-Range Surveillance Applications

Peter Yuen [1,*], Jonathan Piper [2], Catherine Yuen [2] and Mehmet Cakir [1]

1 Centre for Electronic Warfare, Information and Cyber, Defence Academy of the United Kingdom, Cranfield University, Shrivenham SN6 8LA, UK; m.cakir@cranfield.ac.uk
2 Defence Science and Technology Laboratory, Porton Down, Salisbury SP4 0JQ, UK; jpiper@dstl.gov.uk (J.P.); cyuen@dstl.gov.uk (C.Y.)
* Correspondence: p.yuen@cranfield.ac.uk; Tel.: +44-(0)-7766181898

**Abstract:** The fusion of low-spatial-resolution hyperspectral images (LRHSI) with high-spatial-resolution multispectral images (HRMSI) for super-resolution (SR), using coupled non-negative matrix factorization (CNMF), has been widely studied in the past few decades. However, the matching of spectral characteristics between the LRHSI and HRMSI, which is required before they are jointly factorized, has rarely been studied. One objective of this work is to study how the relative spectral response characteristics (R-SRC) of the LRHSI and HRMSI can be better estimated, particularly when the SRC of the latter is unknown. To this end, three variants of enhanced R-SRC algorithms were proposed, and their effectiveness was assessed by applying them for sharpening data using CNMF. The quality of the output was assessed using the L1-norm-error (L1NE) and receiver operating characteristics (ROC) of target detections performed using the adaptive coherent estimator (ACE) algorithm. Experimental results obtained from two subsets of a real scene revealed a two- to three-fold reduction in the reconstruction error when the scenes were sharpened by the proposed R-SRC algorithms, in comparison with Yokoya's original algorithm. Experiments also revealed that a much higher proportion (by one order of magnitude) of small targets of 0.015 occupancy in the LRHSI scene could be detected by the proposed R-SRC methods compared with the baseline algorithm, for an equal false alarm rate. These results may suggest the possibility of SR to allow long-range surveillance using low-cost HSI hardware, particularly when the remaining issues of the occurrence of large reconstruction errors and comparatively higher false alarm rate for 'rare' species in the scene can be understood and resolved in future research.

**Keywords:** super-resolution; coupled non-negative factorization; relative spectral response characteristics; spectral response function; fusion-based super-resolution; CNMF; hyperspectral sharpening

## 1. Introduction

Over the past few decades, hyperspectral imaging (HSI) has been one of the most versatile and effective air- and space-borne remote sensing techniques for the exploration of ground properties, due to the rich spectral content of the imagery [1]. Recent advances in sensor hardware, software, and machine learning tools allow HSI to be widely used across multidisciplinary sectors [2,3], especially for airborne surveillance applications [4]. Despite the very rich spectral information inherent in the HSI, its spatial resolution is substantially lower than its spectral content. This is mainly caused by the relatively large pixel sizes in the focal plane arrays of typical HSI sensors, which are designed to maximize the signal-to-noise ratio (SNR) within the very narrow bandwidth that is received by each spectral band of the sensor. Further constraints, such as the narrow optical entrance slit, typically in the range of 10–100 um for traditional push-broom HSI systems, limit the spatial resolution of the ground image substantially [4]. The low spatial resolution in HSI (LRHSI) has limited

its real usage for long-range surveillance applications, and super-resolution (SR) research, which aims at enhancing the spatial resolution of LRHSI into high-spatial-resolution HSI (HRHSI), has become one of the most popular areas of research within the remote sensing community over the past few decades [5–10]. Amongst the SR research, hardware-based approaches, such as the utilization of hybrid camera systems [11], and sub-pixel shifting (SPS) [12] methods, which de-convolve a series of sub-sampled data to achieve higher spatial resolution images, have also been reported. Although hardware-based SR achieves good performance, the extra capital costs of the imaging hardware mean that the drawbacks generally outweigh their performance benefits. Consequently, the most popular approach to SR research has been software based [5–10], which is capable of recovering the HRHSI scene even from a single input of LRHSI data through the estimation of the blurring kernel in the LRHSI scene [13] (see Equation (2) in Section 2). However, fusion of high-spatial-resolution multispectral images (HRMSI) or high-resolution panchromatic images with low-spatial-resolution hyperspectral images (LRHSI) has been a highly popular and more effective approach for the recovery of HRHSI, with accuracy far better than that using single LRHSI as the input [3,5–10,14–16]. There are a number of fusion schemes designed to implement this kind of SR: the spectral unmixing convex programming approach using coupled non-negative factorization (CNMF) [17–20], various forms of coupled matrix and tensor factorization optimizations (CMTF) [21–23], nonlinear unmixing through intrinsic and extrinsic priors [24], and the implementation of CNMF in deep learning (DL) network architectures [5,25–27], which have been reported in the literature over the past couple of decades.

Although a great deal of work has gone into SR algorithm development, there is very little attention concerning the spectral response characteristic (SRC) mismatch between the LRHSI and HRMSI images, which is required before the pair of datasets can be fused together [18,28,29]. The SRC of an imaging system is specific to its optical and electro-optical configuration. For instance, dependent on the specific imaging hardware systems, the shape of the SRC of a practical sensor can be more Gaussian-like, or it can show varying degrees of asymmetry in its tail; furthermore, the bandwidths of HRMSI are generally broader, to varying degrees, than those of the LRHSI. In practical applications, such as in long-range surveillance operations, the LRHSI and HRMSI image pairs are generally acquired by two different imaging hardware systems. Thus, the relative SRC (R-SRC) between the two image pairs needs to be assessed, such that the raw-image datasets can be adjusted before they are fused. This motivates one of the main themes of the present work, which proposes three different CNMF-based algorithms for enhancing the accuracy of the sharpened scene through several improved R-SRC formulations. This is particularly relevant in this work, as the SRC of the HRMSI data that are employed in this work is not known.

The other objective of the present work is to answer the query from the aerospace industry of whether SR can be utilized for long-range surveillance applications to reduce the need for expensive optics for the HSI hardware. Thus, the second objective of the present work is to assess the utility of the sharpened scene from the target detection perspective. Specifically, we assess whether sub-pixel targets with abundances of about 0.015 in the LRHSI scene can be recovered using SR techniques. More stringent assessment metrics, such as receiver operating characteristics (ROC) and L1-norm-error (L1NE), have been used for assessing the recovery of sub-pixel mixing in the LRHSI for the first time in SR research. Previously, detection of targets of abundance ~0.15 that were 'injected' into the scene synthetically has been performed using anomaly detection [30]. Change detection of large objects with full pixels had also been reported [31]. This paper adopts the more specific adaptive coherent estimator (ACE) [32] as the detector for assessing the recovery of real targets with abundances of about 0.015 in real LRHSI scenes. The present results indicate that conventional assessment metrics, such as spectral angle mapper (SAM) and peak signal-to-noise ratio (PSNR), are not sufficient to quantify the quality of the HRHSI recovery, particularly for the small sub-pixel targets.

The objectives of the present work are based in three areas: (1) To assess the significance of R-SRC between LRHSI and HRMSI for SR of LRHSI using fusion-based techniques. (2) To assess whether the SR technique could be used for long-range surveillance applications without the need for expensive optics and HSI hardware. (3) To assess the limitations of SR, specifically whether a single target pixel with abundances of about 0.015 in the LRHSI scene could be recovered using SR techniques. Based on these objectives, the paper is organized in the following order: Section 1 (this section) outlines the background of SR, the problem statements, and the objectives of the present work. Section 2 presents the proposed CNMF algorithms and compares them with Yokoya's CNMF algorithm [17–19]. Subsequently, the real dataset, which is known as 'Selene' [33], and the assessment metrics that are used in this paper are also presented in Section 2. Section 3 presents the results of the present work, and the conclusion and future outlook of this work are presented in Section 4.

## 2. Methods and Materials

### 2.1. Spectral Unmixing Algorithm

Considering inputs of low-spatial-resolution hyperspectral data ($\mathbf{X}$) and geometrically co-registered high-spatial-resolution multispectral data ($\mathbf{Y}$), which were taken under the same atmospheric and illumination conditions as that of $\mathbf{X}$, the objective of data fusion is to recover the high-spatial-resolution hyperspectral data ($\mathbf{Z}$), such that:

$$
\begin{aligned}
\mathbf{X} &\in R^{L_h \times N_h} \\
\mathbf{Y} &\in R^{L_m \times N_m} \\
\mathbf{Z} &\in R^{L_h \times N_m}
\end{aligned}
\tag{1}
$$

where $L_h$ and $L_m$ denote the number of spectral channels (lambda) in the hyperspectral and multispectral sensors, respectively, and $N_h$ and $N_m$ denote the numbers of pixels in the hyperspectral and multispectral images, respectively. The matrices $\mathbf{X}$, $\mathbf{Y}$, and $\mathbf{Z}$ have the following relationships:

$$\mathbf{X} = \mathbf{ZS} + \varepsilon_s$$

$$\mathbf{Y} = \mathbf{RZ} + \varepsilon_r \tag{2}$$

where $\mathbf{S}$ and $\mathbf{R}$ are the spatial point spread matrix and spectral response matrix between the $\mathbf{X}$ and $\mathbf{Z}$ and the $\mathbf{Y}$ and $\mathbf{Z}$, respectively. The $\varepsilon_s$ and $\varepsilon_r$ are the residue errors between the $\mathbf{X}$ and $\mathbf{ZS}$ and the $\mathbf{Y}$ and $\mathbf{RZ}$, respectively. Note that the sum of each column of $\mathbf{S}$ is normalized to 1 because $\mathbf{X}$ is a spatial linear degradation of $\mathbf{Z}$, and in fact, $\mathbf{S} \in R^{N_m \times N_h}$ can be regarded as a down-sampling blur filter of $\mathbf{Z}$. The matrix $\mathbf{R} \in R^{L_m \times L_h}$ represents the relative spectral responses between the multispectral sensor and the narrow spectral bands of the hyperspectral image $\mathbf{Z}$.

There are many approaches to the recovery of $\mathbf{Z}$, as mentioned in the previous section. Considering the spectral unmixing concept, the objective is to decompose $\mathbf{Z}$ into two components, $\mathbf{E}$ and $\mathbf{A}$, with a minimal unmixing error $\varepsilon_u$:

$$\mathbf{Z} = \mathbf{EA} + \varepsilon_u \tag{3}$$

where $\mathbf{E} \in R^{L_h \times D}$ is the endmember matrix with D endmembers, and $\mathbf{A} \in R^{D \times N_m}$ is the abundance fraction of all endmembers for every pixel in $\mathbf{Y}$. The $\varepsilon_u$ is the fitting error between $\mathbf{Z}$ and $\mathbf{EA}$.

In theory, the matrices $\mathbf{E}$ and $\mathbf{A}$ can be obtained from the decomposition of $\mathbf{X}$ and $\mathbf{Y}$ through the spatially degraded abundance matrix $\mathbf{A}_h = \mathbf{AS}$ (refer to Equation (2)), where $\mathbf{A}_h \in R^{D \times N_h}$, and the spectrally degraded endmember matrix $\mathbf{E}_m = \mathbf{RE}$ (see Equation (2)), where $\mathbf{E}_m \in R^{L_m \times D}$, through the following formulation:

$$\mathbf{X} = \mathbf{EA}_h + \varepsilon_x$$

$$\mathbf{Y} = \mathbf{E}_m \mathbf{A} + \varepsilon_Y$$

and

$$\mathbf{A}_h = \mathbf{AS}$$

$$\mathbf{E}_m = \mathbf{RE} \tag{4}$$

where $\varepsilon_x$ and $\varepsilon_Y$ are the residues for the decomposition of $\mathbf{X}$ and $\mathbf{Y}$, respectively. It can be seen that $\mathbf{E}$ and $\mathbf{A}$ can be readily obtained through alternative unmixing of Equation (4), which forms the basis of the coupled non-negative matrix factorization (CNMF) mechanism. We have adopted Yokoya's CNMF algorithm as the method for unmixing [17–19], but with the addition of several enhancements to the estimation of the relative spectral response characteristics (R-SRC) between $\mathbf{X}$ and $\mathbf{Y}$.

### 2.2. Relative Spectral Response Characteristic (R-SRC) Estimation

Due to the different spectral response characteristics (SRC) between the low-spatial-resolution HSI, $\mathbf{X}$, and the high-spatial-resolution MSI, $\mathbf{Y}$, which are commonly taken by very different imaging hardware, the relative spectral response characteristics (R-SRC) of the two datasets must be estimated, such that consistent spectral content between them is established before the coupled matrix decomposition can take place.

In order to establish the spectral response function between the LRHSI, $\mathbf{X}$, and the HRMSI, $\mathbf{Y}$, it is necessary to reduce the resolution of $\mathbf{Y}$ to pixel match with that of $\mathbf{X}$.

Consider Equation (2), which can be approximated as $\mathbf{Y} \cong \mathbf{RZ}$; thus,

$$\mathbf{YS} \cong \mathbf{RZS} = \mathbf{RX}$$

We then write:

$$\mathbf{Y}_h = \mathbf{YS}$$

Thus,

$$\mathbf{Y}_h = \mathbf{RX} \tag{5}$$

$\mathbf{Y_h} \in \mathbf{R}^{L_m \times N_h}$, and it represents the low-spatial-resolution version of the original high-spatial-resolution multispectral image $\mathbf{Y}$. The purpose of $\mathbf{Y_h}$ is the formation of a version of $\mathbf{Y}$ with degraded spatial resolution, such that it is pixel matched with the LRHSI, to allow the spectral responses, R, between these two sets of data to be deduced over the entire scene. Note that this is the first level of spectral matching between the LRHSI and the HRMSI. Equation (5) implies that the relative spectral response characteristic (R-SRC) can be estimated through the mutual overlapping of spectral bands in $\mathbf{X}$ and $\mathbf{Y_h}$. Therefore, the problem requires that, for each multispectral band, $i$, we obtain $r_i$ to solve the following optimization problem:

$$arg \, \underset{r}{min} \, \parallel \mathbf{Y}_{h_i} - r_i \mathbf{X} \parallel_2^2$$

$$subject \, to \, \left| r_i - r_i' \right| \leq r_i'.\varepsilon, \, and \, r_i \geq 0 \tag{6}$$

where $\mathbf{Y}_{h_i} \in \mathbf{R}^{1 \times N_h}$ and $r_i \in \mathbf{R}^{1 \times L_h}$ are the column vectors of $\mathbf{Y_h}$ and $\mathbf{R}$, respectively, $\varepsilon$ is the preset small positive threshold, and $r_i'$ is the $r_i$ row vector of the initial estimate of $R$. The initial value of $r_i'$ is a zeros vector with ones at the wavelength of $\mathbf{Y}$.

Note that Equation (6) optimizes $r_i$ band-by-band ($i = 1, 2, \ldots, L_m$), which is different from Lanaras's algorithm [28,34], which optimizes a small subset (~6) of bands in each step of the optimization process. The number of bands in the subset that Lanaras employed was obtained through prior information of spectral characteristics of $\mathbf{X}$ and $\mathbf{Y}$. Since this kind of prior information is not readily available in general practice, we prefer to adopt Yokoya's formulation in the current work to optimize $r_i$ band-by-band.

### 2.2.1. R-SRC of Yokoya's Original CNMF Algorithm (Y-CNMF)

Yokoya's original algorithm [17–19] utilized MATLAB's quadratic convex optimization programming code (CVX) for the estimation of $\mathbf{R}$ using Equation (6), with $\varepsilon = 0$ and maximum iterations of 1500. Note that the optimized $\mathbf{R}$ yielded by Equation (6) is the difference of spectral responses between the mutually overlapping spectral bands of $\mathbf{X}$ and $\mathbf{Y_h}$; thus, $\mathbf{Y}$ is corrected after $\mathbf{R}$ is obtained from Equation (6) before the CNMF process takes place:

$$Y_h = \begin{cases} Y_h - RX & if\ Y_h > RX \\ \mathbf{0} & if\ Y_h < RX \end{cases} \tag{7}$$

Yokoya's original algorithm forces $\mathbf{Y}$ to be positive through the second condition of Equation (7). It was found that this ad-hoc condition induces low-radiance pixels (i.e., when $y_{hi} < r_i X$) which become zeros in radiance, making them appear as 'black' pixels in the scene. Experiments also revealed that the radiance of pixels in the sharpened scene obtained from the Y-CNMF method deviated significantly from the ground truth (see Section 3).

### 2.2.2. Enhanced R-SRC Algorithm (E-CNMF)

Rather than forcing $y_{hi} = 0$ when $r_i X > y_{hi}$, as in Equation (7), the min($\mathbf{RX}$) is subtracted from $\mathbf{Y_h}$, such that all $y_{hi} > 0$:

$$Y_h = \begin{cases} Y_h - RX & if\ Y_h > RX \\ Y_h - min(RX) & if\ Y_h < RX \end{cases} \tag{8}$$

Equation (8) maintains the positivity of $\mathbf{Y}$ after it is corrected by min($\mathbf{R}$) instead of forcing them to zero, as in Equation (7), when $y_{hi} < r_i X$. This measure tends to reduce the number of overcorrected pixels that are seen from the sharpened scene produced by the Y-CNMF algorithm. Although the enhanced version, E-CNMF, improves the quality of the sharpened image to some extent, it is also an ad-hoc formulation without sound underlying principles.

### 2.2.3. Constrained Enhanced R-SRC Algorithm (CE-CNMF)

Equation (6) attempts to minimize the L2 norm of the differences between $\mathbf{Y_h}$ and $\mathbf{RX}$, and the $r_i$ of $\mathbf{R}$ can be further constrained within a lower and upper bound during the optimization cycles. Since $\mathbf{R}$ represents the relative spectral response between the $\mathbf{X}$ and $\mathbf{Y_h}$, the following constraint is applied in this version of the algorithm:

$$arg\ {min \atop r}\ \parallel Y_{h_i} - r_i X \parallel_2^2$$

$$subject\ to\ \left| r_i - r_i' \right| \leq\ r_i'.\ \varepsilon\ ,\ and\ 1 \geq r_i \geq 0 \tag{9}$$

The $r_i$ is bounded within [0, 1] in this case, and the resulting $\mathbf{Y}$ is then obtained by applying Equation (8) before it is passed over to the CNMF processing stage.

### 2.2.4. Second-Level CNMF with Constrained Enhanced R-SRC Algorithm (CEY-CNMF)

It is well known that techniques for learning perception through its parts, such as matching pursuit or matrix factorization alike, suffer drawbacks of not being able to recover highly articulated objects or databases. Learning parts from highly complex data requires hierarchical models that contain multiple levels of hidden variables, rather than just using a single layer of parts in the learning process [35].

One objective of this work was to assess the capability of parts' learning techniques, such as NMF, and particularly, whether CNMF is capable of recovering small target pixels with a small occupancy of about 0.015. To achieve this objective, the CNMF process was cascaded into two different levels, with the first one dealing with raw $\mathbf{X}$ and $\mathbf{Y}$ using the CE-CNMF, as set out in Section 2.2.3 above. The number of columns in $\mathbf{E}$ in Equation (4),

as determined by the vertex component analysis (VCA) algorithm [36], was small due to the small number of pixels ($N_h$) and the coarse spectral fidelity in the raw data **X**, which typically yielded in the order of <40 endmembers in the test scenes that were used in this work. The sharpened result of the first round (**X′**) contained a substantially larger number of pixels ($N_m$) with substantially higher spectral fidelity, which induced a larger number of **E**, typically about 3–4 times higher than that of the first round. Thus, the CEY-CNMF algorithm was deployed as follows:

$$\text{Input } [\textbf{X, Y}] \rightarrow \text{CE-CNMF} \rightarrow \text{1st stage output } [\textbf{X′}] \rightarrow \text{2nd stage CNMF input } [\textbf{X′,Y}] \rightarrow \text{Y-CNMF} \rightarrow \text{output } \textbf{X″} \quad (10)$$

where **X′** and **X″** are the sharpened output of **X** in the first and second rounds of CNMF processing, respectively. Note that the Y-CNMF algorithm was adopted in the second round because the R-SRC between the [**X′,Y**] was negligibly small, unlike in the first round, where the difference in SRC in [**X, Y**] was significantly larger.

*2.3. Implementation of CNMF for Spectral Unmixing: Second Level of Spectral Matching between LRHSI and HRMSI*

We considered a non-negative matrix **Z**, as in Equation (3), with the objective of finding non-negative factors, **E** and **A**:

$$\textbf{Z} \cong \textbf{EA} \quad (11)$$

where **E** is the basis, which can be optimized to approximately realize **Z** through a linear combination with the sparse matrix **A**. The cost function for achieving the objective is to minimize the distance or, alternatively, the divergence between **Z** and **EA:**

$$arg \, {min \atop E,A} \parallel \textbf{Z} - \textbf{EA} \parallel_2^2 \quad (12)$$

Lee and Seung [35,37] demonstrated that the update for minimizing the distance between **Z** and **EA** is much more computationally efficient using a multiplicative factor than using conventional additive gradient descents for the decomposition of **Z**:

$$A \leftarrow A \left( \frac{E^T Z}{E^T E A} \right) \quad and \quad E \leftarrow E \left( \frac{Z \, A^T}{E \, A A^T} \right) \quad (13)$$

where $(.)^T$ denotes the transpose of the matrix. Note that the multiplicative update rule in Equation (13) is designed for the decomposition of a single matrix, **Z**. Equation (4) indicates that the sharpening of low-spatial-resolution HSI (**X**) requires the cueing of high-spatial-resolution MSI (**Y**), and Equation (4) can be approximated as:

$$\textbf{X} \cong \textbf{EA}_h$$

$$\textbf{Y} \cong \textbf{E}_m\textbf{A} \quad (14)$$

Thus, the coupled decomposition of **X** and **Y** is needed for solving our current problem, and the update rule for the CNMF in Equation (14) becomes:

For the decomposition of **X**:

$$arg \, {min \atop E,A} \parallel \textbf{X} - \textbf{EA}_h \parallel_2^2$$

$$E \leftarrow E. * \left( \frac{X \, A_h^T}{E A_h A_h^T} \right) \quad and \quad A_h \leftarrow A_h. * \left( \frac{E^T X}{E^T E A_h} \right) \quad (15a)$$

For the decomposition of **Y:**

$$arg \, {min \atop E,A} \parallel \textbf{Y} - \textbf{E}_m\textbf{A} \parallel_2^2$$

$$E_m \leftarrow E_m.* \left( \frac{Y\,A^T}{E_m\,AA^T} \right) \quad and \quad A \leftarrow A.* \left( \frac{E_m^T Y}{E_m^T E_m A} \right) \tag{15b}$$

The (.*) and (/) denote elementwise multiplication and division in Equation (15).

CNMF begins with the NMF decomposition of the low-spatial-resolution HSI **X**, according to Equation (15a), due to its richness in the spectral content of **X**. This is needed in order to establish the initial values for both **E** and $\mathbf{A_h}$ before the coupled decomposition of **X** and **Y** can proceed. Firstly, $\mathbf{A_h}$ is obtained through the second part of Equation (15a) by using $1/D$ as the initial values for $\mathbf{A_h}$, where D is deduced from **X** using vertex component analysis (VCA) [36], and **E** is kept constant until $\mathbf{A_h}$ converges. As soon as the initial $\mathbf{A_h}$ is obtained, both **E** and $\mathbf{A_h}$ are alternatively updated according to Equation (15a) until convergence. Thus, this initialization step outputs the initial values of **E** and $\mathbf{A_h}$, which are needed for the decomposition of **X** and **Y** in the following coupling cycles.

The coupled decomposition begins with the factorization of the high-spatial-resolution **Y** in the same manner as that for the decomposition of **X**: the **A** is obtained through the second part of Equation (15b) by using $1/D$ as initial values for **A**, and $\mathbf{E_m}$ is kept constant until **A** converges. As soon as the initial **A** is obtained, both $\mathbf{E_m}$ and **A** are alternatively updated according to Equation (15b) until convergence. Note that the initial values of **E**, $\mathbf{E_m}$, $\mathbf{A_h}$, and **A** can be obtained up to this point.

The next stage of the coupled decomposition is the factorization of **X** in two steps: firstly, to update **E** by keeping $\mathbf{A_h}$ constant through Equation (15a), then to optimize both **E** and $\mathbf{A_h}$ through Equation (15a,b). This iterative updating forms the inner loop of the overall coupled decomposition procedure.

These nested loops of NMF unmixing are repeated until the number of iterations exceeds a user-defined total, or the RMSE (of **E**, $\mathbf{E_m}$, $\mathbf{A_h}$, and **A**) in the current loop with respect to the previous one exceeds a user-defined threshold, whichever occurs earlier. The short code of the proposed algorithms is presented in Algorithm 1 below.

*2.4. HSI Datasets*

Two subsets of the Selene hyperspectral data [33] were used as the ground truth (GT) for assessing the effectiveness of the sharpening algorithms in this work. The scene was acquired by the HySpex VNIR-1600 hyperspectral sensor over the Porton Down Range (Long 51°8′19.7″ N, Lat 1°39′16.9″ W to 51°7′41.7″ N 1°40′8.5″ W) on 12 August 2014 BST 12:00:04 at an altitude of 4000 ft, with a ground sampling distance (GSD) of about 0.5 m. The scene contains 95% natural materials, such as grass, soil, and trees, with artificial manmade materials, such as ground markers, path, concrete, buildings, and various sizes of colored panels, for calibration and target detection purposes, as the remaining materials in the scene. The scene was radiometrically and atmospherically corrected using QUAC [38], and two subsets of the scene were selected as the ground truth (GT) high-resolution HSI (HRHSI) datasets for the present sharpening work. Subset 0 consists of 7 colored manmade calibration panels in sizes of $2 \times 2$ m, with backgrounds of bushes, grassland, buildings, and paths. Subset 1 contains mainly grassland but is characterized by having very small ceramic tiles as targets in sizes of $0.2 \times 0.2$ m scattered around the scene. These two subsets of the scene were then linearly aggregated to a GSD of 3 m to create the LRHSI used in this study. The pixel occupancy of the green ceramic tile targets in the aggregated subset 1 scene was about 0.015. Please refer to [4,33] for more information about the properties of the datasets, such as the experimental settings, design of experiment, control variables, and data selections. The pseudo-RGB of these two subsets and their aggregated scenes are depicted in Figure 1, and the target map of the small ceramic tiles is shown in Figure 2.

---

**Algorithm 1.** The short code for all algorithm utilized in the present work

---

*Inputs: low-spatial-resolution hyperspectral data, $X \in R^{L_h \times N_h}$; high-spatial-resolution multispectral data, $Y \in R^{L_m \times N_m}$; threshold inner loop, $\varepsilon_1 = 1 \times 10^{-8}$; threshold outer loop, $\varepsilon_2 = 1 \times 10^{-2}$; threshold max N of inner loop, $N_1 = 200$; threshold max N of outer loop, $N_2 = 1$; max N of loops for R-SRC estimation, $N_3 = 1500$.*
*Outputs: endmember, $E \in R^{L_h \times D}$, with D endmembers; abundance matrix, $A \in R^{D \times N_m}$*

A      Estimate R-SRC between **X** and **Y** using MATLAB 'quadprog' command with max $N_3$ loops

1: Y-CNMF: adjust **Y** using Equation (7)
2: E-CNMF: adjust **Y** using Equation (8)
3: CE-CNMF: adjust **Y** using Equation (9)
4: CEY-CNMF: adjust **Y** and cascade end results for second level of NMF, as shown in the flow chart in (10).

B      CNMF for the decomposition of **X** and **Y** into **E** and **A**

(i)     Initialization Step: Obtain the initial **E** and $\mathbf{A_h}$ through decomposition of **X**:
Obtain D by VCA, with constant **E** to update $\mathbf{A_h}$ through Equation (15a) first, then optimize both **E** and $\mathbf{A_h}$ by Equation (15a).

(ii)    CNMF Outer loop:
Begin
Decomposition of **Y**
(a) Use the **E** that was obtained in (i) to set $\mathbf{E_m}$ through Equation (4), then **A** is updated with constant $\mathbf{E_m}$ through Equation (15b).
(b) Once the initial **A** is obtained, both $\mathbf{E_m}$ and **A** are updated through Equation (15b).
(c) Loop if the differences in ratios of RMSE($\mathbf{E_m}$) and RMSE(**A**) w.r.t. the previous loop are greater than $\varepsilon_2$ or the maximum number of loops is less than $N_2$.

(iii)   CNMF Inner loop:
Begin
BeginDecomposition of **X**
(a) Use the **A** that was obtained from (ii) to set $\mathbf{A_h}$ through Equation (4), then **E** is updated with constant $\mathbf{A_h}$ through Equation (15b).
(b) Once the initial **E** is obtained, both **E** and
$A_h$ are updated through Equation (15b).
(c) Loop if the differences in ratios of RMSE(**E**) and
RMSE($\mathbf{A_h}$) w.r.t. the previous loop are greater than $\varepsilon_1$ or the maximum number of loops is less than $N_1$.
End;

      End.

---

The HRMSI was acquired at the same time as HSI acquisition by using a mega-pixel RGB camera. The modulation transfer function (MTF) and the sensitivity of the mega-pixel camera are not known. This study may serve as a test to verify the proposed SR algorithms, regarding whether the enhanced R-SRC algorithm for super-resolution (SR) using CNMF is capable of recovering sub-pixel targets at an abundance of ~0.015 while the SRC of the HRMSI is unknown.

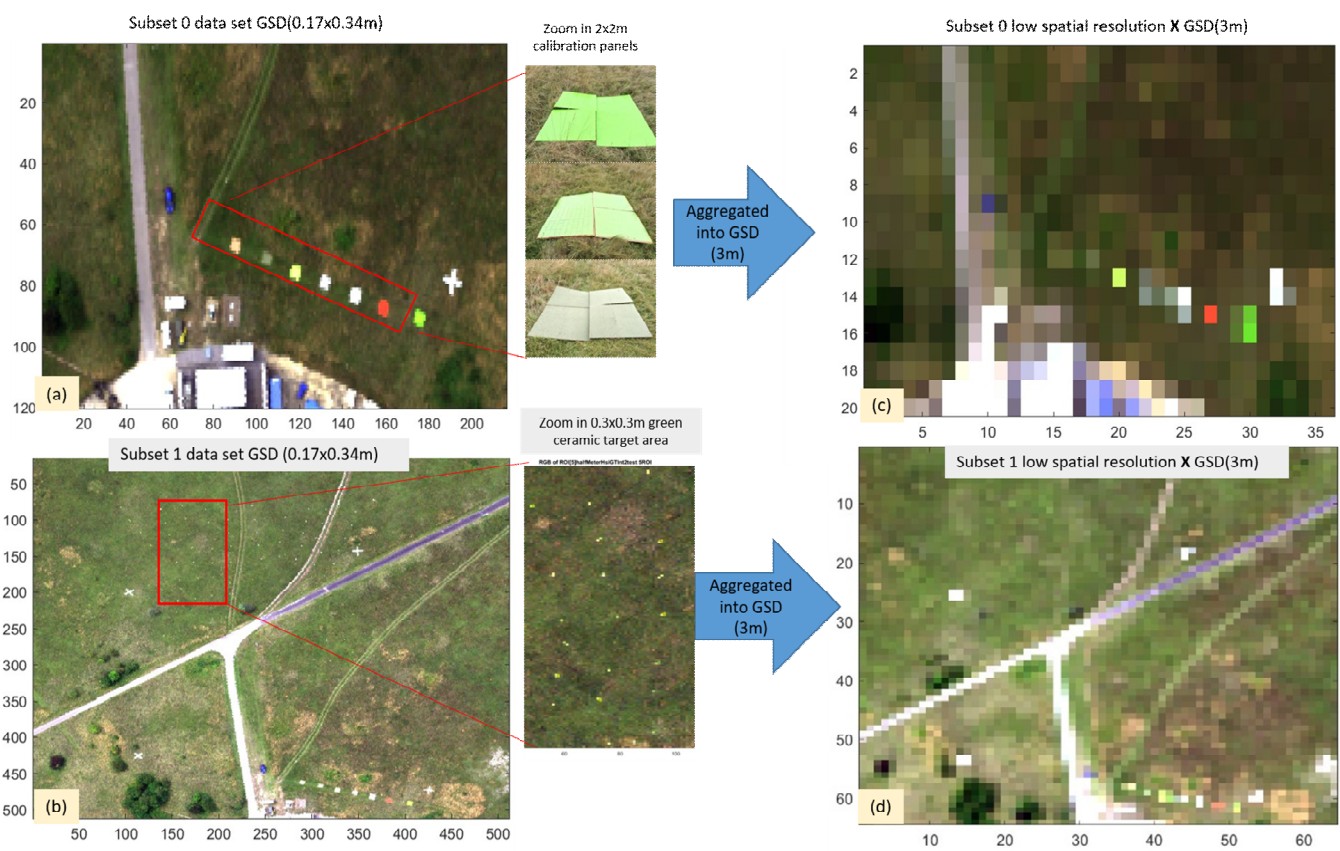

**Figure 1.** The pseudo-RGB of the GT HSI scene, with GSD 0.17 × 0.34 m for (**a**) subset 0 and (**b**) subset 1, and their corresponding aggregated scenes (**c**,**d**), respectively, at GSD of ~3 m. The pixel occupancy of the green ceramic tiles as targets in (**d**) was about 0.015.

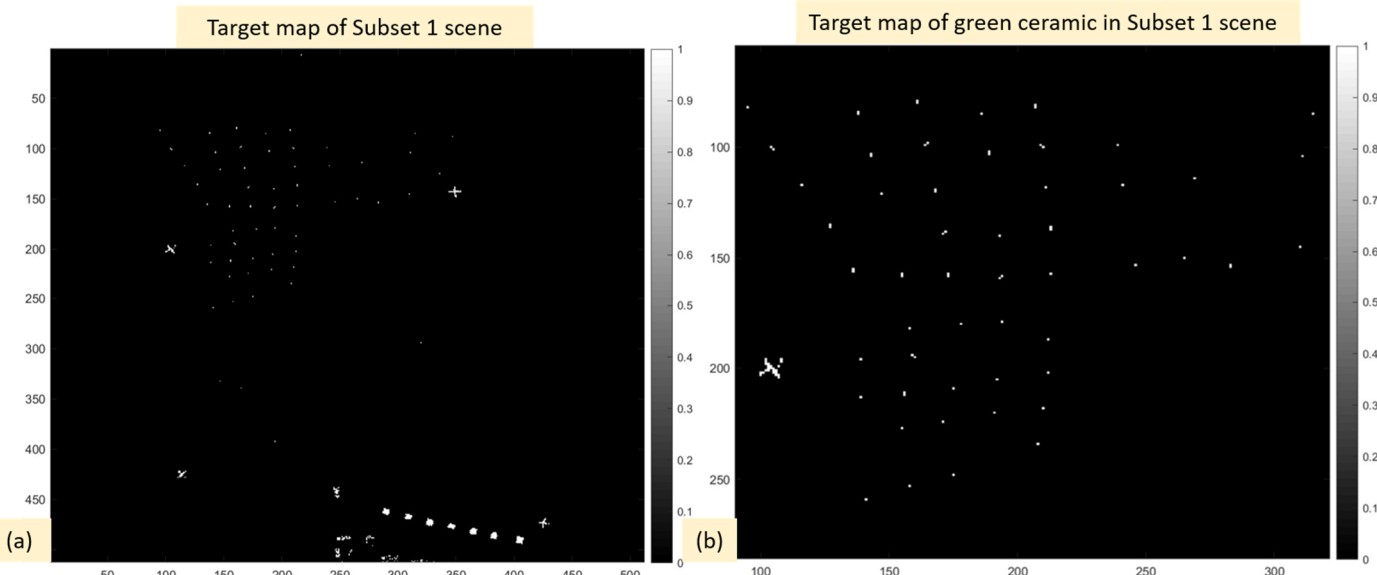

**Figure 2.** The manmade materials as targets in the subset 1 scene: (**a**) the seven calibration panels and green ceramic tiles in the entire scene, and (**b**) the area where small green ceramic tiles are located in the subset 1 scene.

*2.5. Assessment of the Effectiveness of Spectral-Unmixing-Based SR*

The assessment metrics that were adopted in this study were the L1-norm-error (L1NE), adaptive coherent estimator (ACE) [32], receiver operating characteristics (ROC) [39,40], and the spectral angle mapper (SAM) [40] for a direct comparison with L1NE:

$$\text{L1NE}(\text{SR}, \text{GT}) = \left| \frac{\ell_1^{GT} - \ell_1^{SR}}{\ell_1^{GT}} \right| \times 100\%$$

$$\text{ACE}(\text{S}, \text{x}) = \frac{\left| S^t \Gamma^{-1} x \right|^2}{\left( S^t \Gamma^{-1} S \right) \left( x^t \Gamma^{-1} x \right)}$$

$$\text{SAM}(\text{S}, \text{x}) = \frac{1}{n} \sum_{j=1}^{n} arccos \left[ \frac{S_j^t . x_j}{\left| S_j^t \right| . \left| x_j \right|} \right] \tag{16}$$

where $\ell_1^{GT}$ and $\ell_1^{SR}$ represent the L1 norm of the scene pixels, of the ground truth and the SR sharpened scenes, respectively, $S^t$ is the transpose of the reference spectrum (e.g., target spectral signature for detection), $\Gamma$ is the covariance of the background pixels of the scene, and x is the test pixel.

## 3. Super-Resolution (SR) Results

*3.1. Significance of R-SRC for the Effectiveness of SR*

To understand the significance of R-SRC, particularly when the fusion approach is used for SR, this section presents the sharpening result for the LRHSI dataset 0 by fusing it with two different versions of the HRMSI: one is the raw RGB version (RGB-HRMSI), and the other is the reduced normalized rgb (r-rgb-HRMSI) of the raw RGB HRMSI:

$$\text{r-rgb} : \text{f}(\overline{\overline{r}}, \overline{\overline{g}}, \overline{\overline{b}}) = \left( \frac{(R, G, B)}{R + G + B} \right) \tag{17}$$

where R, G, and B represent the radiance intensity of the red, green, and blue channels of the RGB HRMSI, and $\overline{\overline{r}}$, $\overline{\overline{g}}$, $\overline{\overline{b}}$ represent the normalized radiance of the red, green, and blue channels in the r-rgb-HRMSI.

Figure 3 depicts the SR of the aggregated dataset 0 LRHSI through CNMF with two different versions of HRMSI, namely the three-band RGB-HRMSI and the reduced normalized r-rgb-HRMSI, by using the same Y-CNMF algorithm with all algorithm parameter the same in both cases. The averaged L1NE error maps of these two SR scenes are seen to be distinctly different: the one that was fused with RGB-HRMSI resulted in an average scene error of ~24%, which is almost double that of the results produced using the reduced normalized r-rgb-HRMSI. This result may represent the **first** evidence to show the significance of R-SRC for spectral matching/calibration between the LRHSI and the HRMSI dataset pairs in fusion-based SR work.

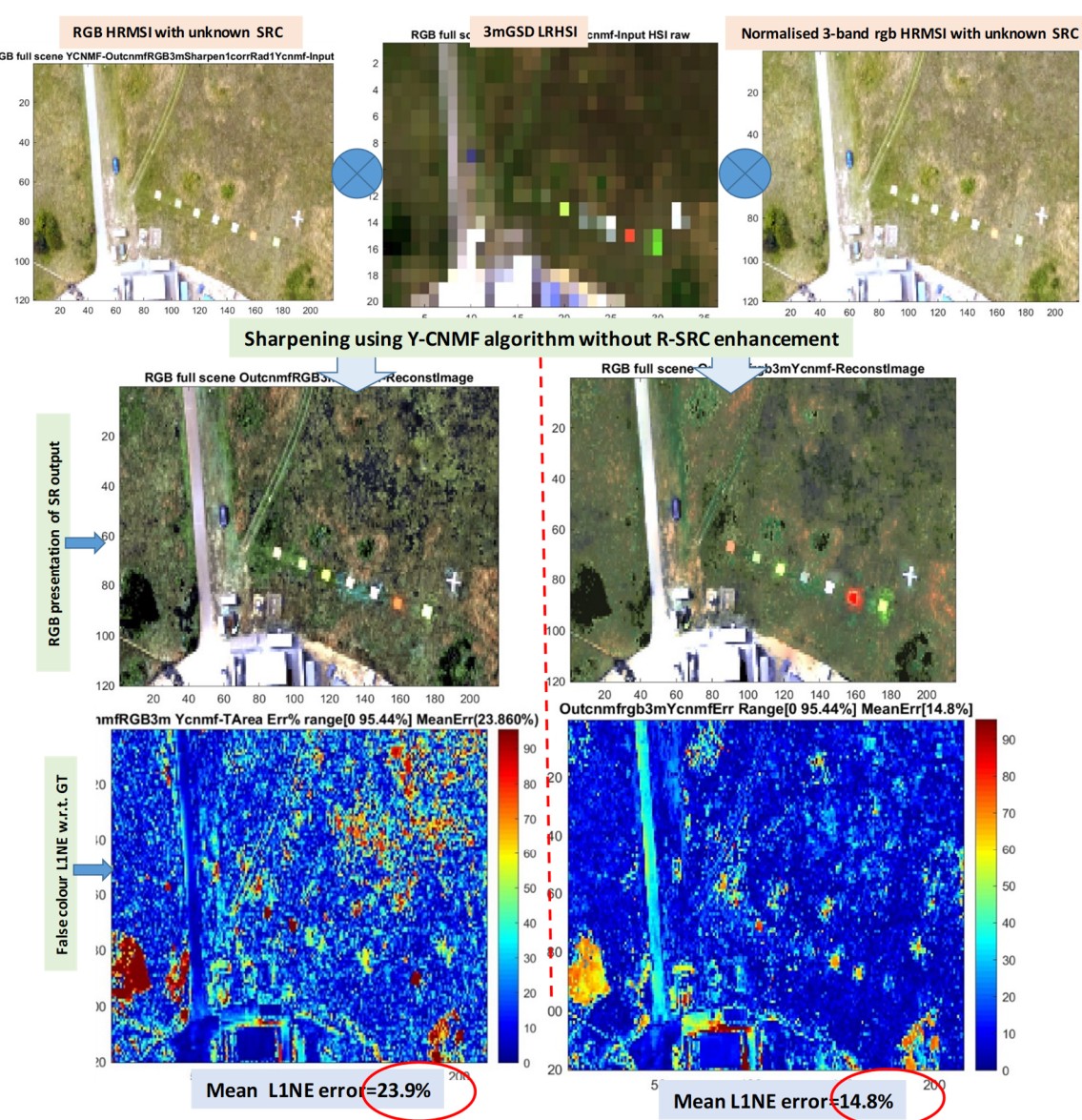

**Figure 3.** The significance of R-SRF, highlighting the differences in SR accuracy when the 3 m GSD LRHSI (**middle top**) was fused with two versions of the HRMSI: (**Left**) the RGB version of HRMSI and (**right**) the normalized rgb of HRMSI. In both cases, the same algorithm (Y-CNMF) was used for SR, but the L1NE error of the RHS result was half that of the left.

*3.2. Subset 0: SR Performance of the Proposed E-CNMF, CE-CNMF, and CEY-CNMF vs. Y-CNMF*

3.2.1. Performance of SR Assessed by L1NE vs. SAM

Figures 4 and 5 present the L1NE and SAM assessments of SR outputs from the proposed SR algorithms and compare the sharpening performance with results produced by Yokoya's original algorithm (Y-CNMF). The inputs for the coupled decomposition were the 3 m GSD LRHSI and the three-band RGB-HRMSI datasets. It is seen from Figure 4 that the three proposed SR algorithms exhibited a ~3 times smaller average L1NE for the whole scene than Y-CNMF. Recall that the only difference between the three proposed SR algorithms w.r.t. the Y-CNMF is the implementation of various enhancements in the R-SRC estimation. Thus, the result here may provide the second piece of evidence to demonstrate the significance of the R-SRC estimation in the fusion-based SR process, which requires appropriate spectral matching between the two input datasets (LRHSI and HRMSI).

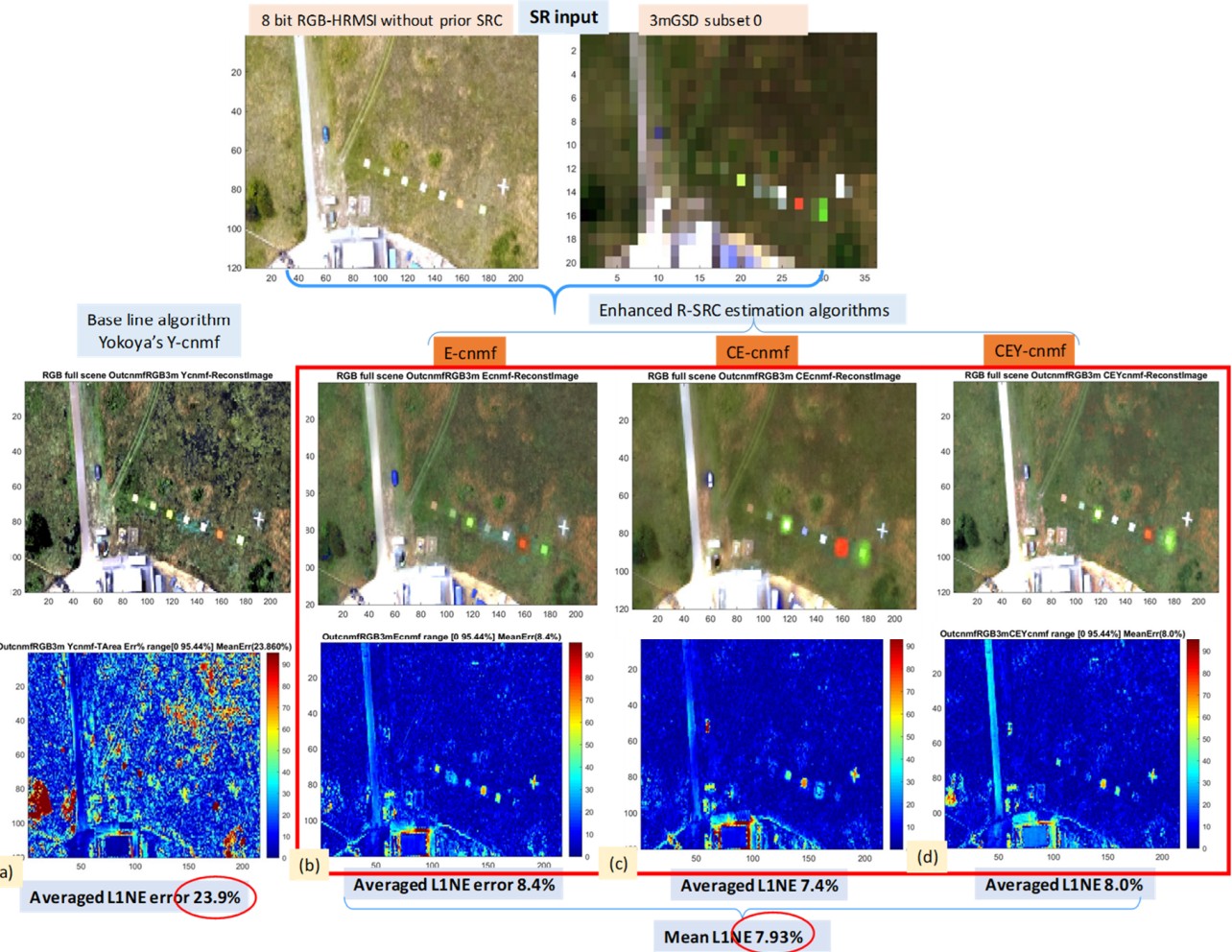

**Figure 4.** The performances of SR through the fusion of the 3 m GSD LRHSI of subset 0 with the RGB-HRMSI using all SR algorithms proposed in this work. Top panel: RGB representation of input datasets. Middle panels (**a–d**): the RGB representations of the outputs of the four SR algorithms: Yokoya's original algorithm Y-CNMF, E-CNMF, CE-CNMF, and CEY-CNMF, respectively. Bottom panel: the corresponding L1NE w.r.t. the GT for the SR outputs obtained from the four SR algorithms. It can be seen that the proposed algorithms (**b–d**) exhibited a factor of ~3-fold of error reduction compared with Yokoya's original algorithm, shown in (**a**).

It is of interest to present the results of Figure 5 using SAM, which has been one of the most popular assessment metrics used in SR research over the last few decades [1–33]. Figure 5 presents the SAM results for the three SR outputs of the proposed SR algorithms, together with those produced for Yokoya's original algorithm (Y-CNMF). All are presented in false color maps. The mean of the SAM over the entire scene for the three proposed SR algorithms was about 0.058, which is ~22% better than that of Yokoya's algorithm (Y-CNMF), while the L1NE assessment that is shown in Figure 4 indicated a ~3-fold reduction of spectral errors for the SR reconstructions by the proposed algorithms. The significant difference between these two assessments may suggest the inadequacy of SAM for assessing the accuracy of SR results, particularly when SR is intended for surveillance or target detection applications (see Sections 3.2.2, 3.3.2, and 3.3.3 for more information).

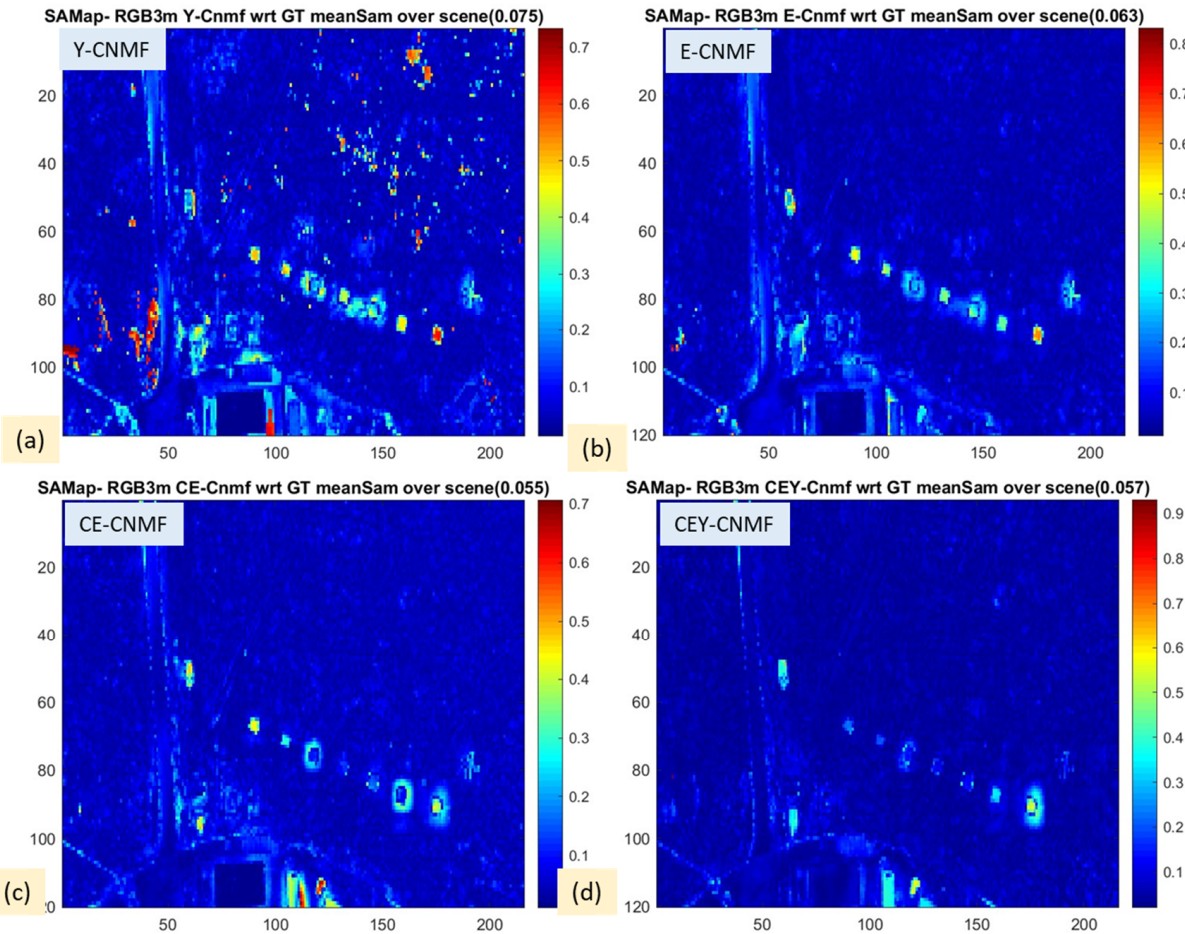

**Figure 5.** The false color maps of the SAM results for the SR of the Selene subset 0 scene processed by (**a**) Y-CNMF, (**b**) E-CNMF, (**c**) CE-CNMF, and (**d**) CEY-CNMF algorithms. The average SAM over the entire scene for the three proposed algorithms in (**b–d**) was seen to exhibit only a ~20% improvement over the original Y-CNMF, while the L1NE assessment that is shown in Figure 4 indicated a ~3-fold reduction in spectral errors for the SR reconstructions by the proposed algorithms.

Since SAM only measures the angle between the spectral vectors, while the norm of vectors is dependent on the absolute radiance (or reflectance) of pixel spectra, the L1NE metric is expected to be more appropriate for assessing the SR scenes, particularly for detection of specific target applications, where the absolute magnitude of a pixel vector is important, too. Figure 6 plots the mean pixel spectra within small regions of interest (ROIs) of the sharpened subset 0 scenes generated by all four SR algorithms. It can be noted that the magnitudes of pixel spectra of the scene sharpened using Yokoya's method (plotted in red dots) deviated rather significantly from the GT, although the shapes of spectra were similar. Contrastingly, the spectra of the sharpened scenes processed by the proposed R-SRC enhanced SR algorithms were found to have closer spectral resemblance to the GT.

The results presented in Figures 4 and 6 may further confirm the significance of the R-SRC estimation, which serves as an effective spectral calibration between the two input datasets, LRHSI and HRMSI, particularly when they are acquired by completely different imaging hardware. A good spectral matching between the two input datasets is fundamentally important in the fusion-based SR process to ensure the recovery of both spatial and spectral characteristics of the HRHSI. The spatial and spectral integrities of the sharpened scene are the most fundamental requirement for applications, such as in long-range surveillance (see Sections 3.2.2 and 3.3.3 below).

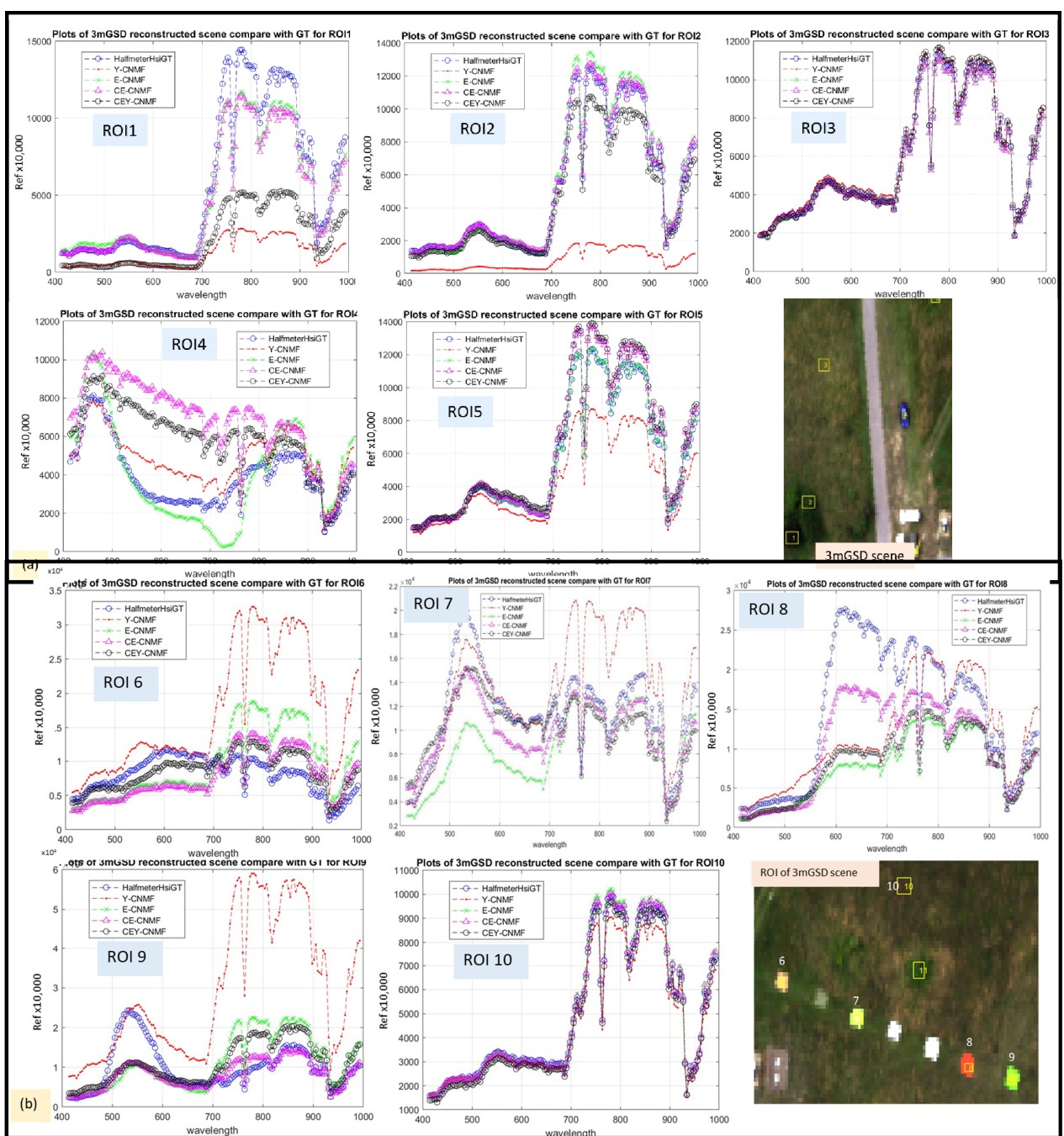

**Figure 6.** Selection of pixel spectra of the subset 0 scene sharpened by all SR algorithms: (**a**) natural vegetation and (**b**) manmade materials. The scene sharpened by Yokoya's algorithm is plotted in red dots, which are seen to deviate greatly from the ground truth (GT; plotted in blue circles). It can be seen that the proposed algorithms, particularly CEY-CNMF (plotted in magenta triangles), performed much better than Yokoya's algorithm, with substantially closer spectral resemblance to the GT. The location of the ROI are depicted in the RGB picture of the scene.

### 3.2.2. ROC Assessment for the Recovery of Large Targets with Abundance 0.44

One objective of the present work was to examine whether SR techniques may allow long-range surveillance or target detection to function satisfactorily, even when using a low-cost LRHSI system together with mega-pixel RGB HRMSI hardware. The requirement hinges on assessing the limits of current state-of-the-art machine learning and software

capabilities, with a view to understanding where the technology gap lies and, subsequently, to allowing the derivation of future plans to realize the objective via practical field trials.

Figure 7 plots the ROC for the detection of several 2 × 2 m calibration panels in subset 0 after the 3 m GSD LRHSI is sharpened by the proposed algorithms. The abundance of the panels in the LRHSI is about 0.44, and the target signatures of the panels were extracted from the GT HRHSI dataset for detection by the ACE (Equation (16)) algorithm. It is seen from the figure that the detection of these relatively large targets in scenes that were sharpened by the proposed enhanced R-SRC CNMF algorithms was far better than what was shown by Yokoya's original SR algorithm (in the red dot plot). Some panels, such as the target in ROI(1), exhibited a three orders of magnitude false alarm rate reduction, for the same detection rate, compared with Yokoya's algorithm. The better detection of targets from the scene indicates a more accurate reconstruction of the HRHSI.

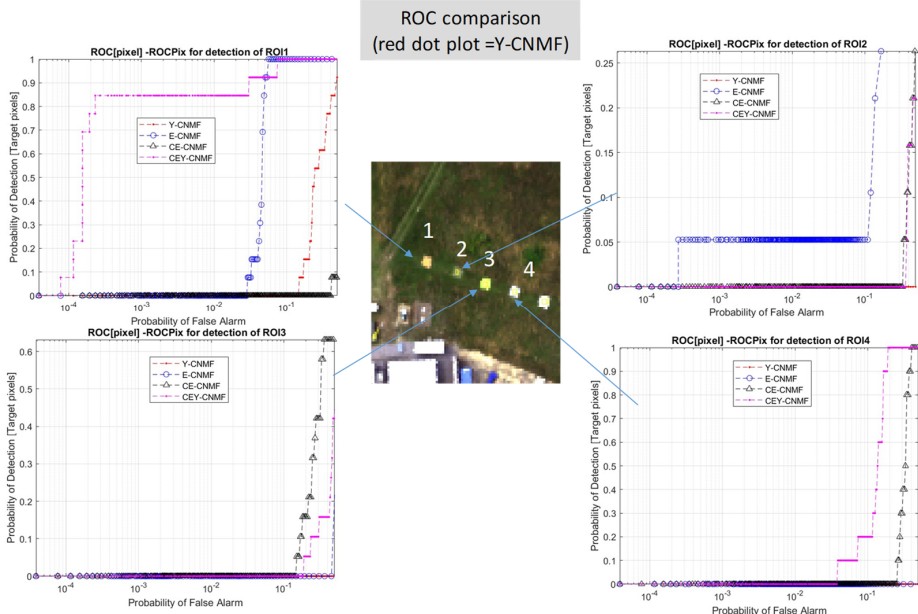

**Figure 7.** The ROC plots for the detection of four manmade panels by the ACE detector from the sharpened subset 0 scene processed by the proposed algorithms, compared with ROC curves for Yokoya's original algorithm (in the red dot plot). It is seen that the proposed algorithms achieved several orders of magnitudes better false alarm reduction compared with Yokoya's algorithm for most targets in the scene.

This result is consistent with the conclusion from Section 3.2.1, that the enhanced R-SRC calibration between the LRHSI and the HRMSI proposed in this work is an effective tool for facilitating the recovery of HRMSI with high spectral and spatial integrity, particularly when the fusion approach is adopted for super-resolution.

### 3.3. Subset 1: SR Performance of the Proposed E-CNMF, CE-CNMF, and CEY-CNMF vs. Y-CNMF

Another objective of the present work was to investigate the limits of SR to examine its capability for the recovery of small targets when they were strongly mixed with the background in the scene. For this purpose, subset 1 of the Selene scene was selected for experimentation. The scene contained both large calibration panels of sizes 2 × 2 m, as well as a number of small green colored ceramic targets of dimensions 0.2 × 0.2 m. The RGB representation of the scene is shown in Figure 8, and a zoomed-in image of the single-pixel small green ceramic targets is depicted in the inset. The GSD of the HRHSI GT was ~0.5 m, which was then linearly aggregated into GSD 3.2 m to form the LRHSI as the test dataset. The RGB representations of the subset 1 LRHSI and its corresponding HRMSI (at GSD of 0.5 m) scenes are presented in Figure 9.

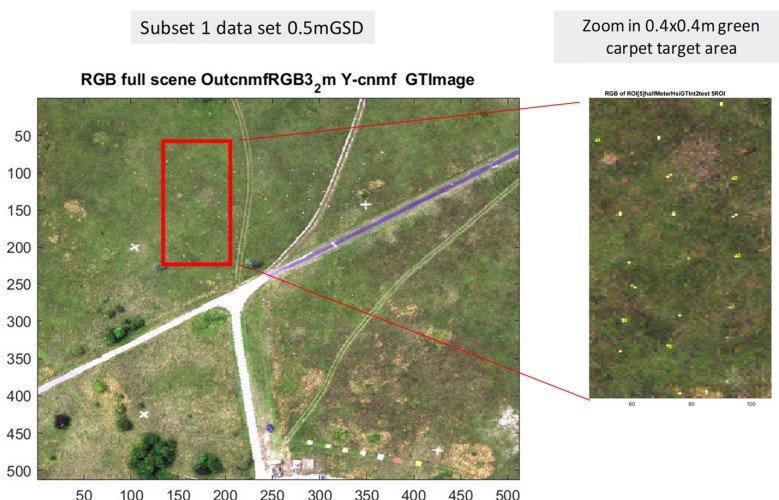

**Figure 8.** The RGB representation of subset 1 (0.5 m GSD), which was embedded with small camouflage targets (green ceramic) of ~1 pixel in size (0.2 × 0.2 m) in the scene. The purpose of this dataset was to assess the limitations of the sharpening algorithms, especially considering whether they can recover single-pixel targets from the 3 m GSD LRHSI scene with abundances of ~0.015.

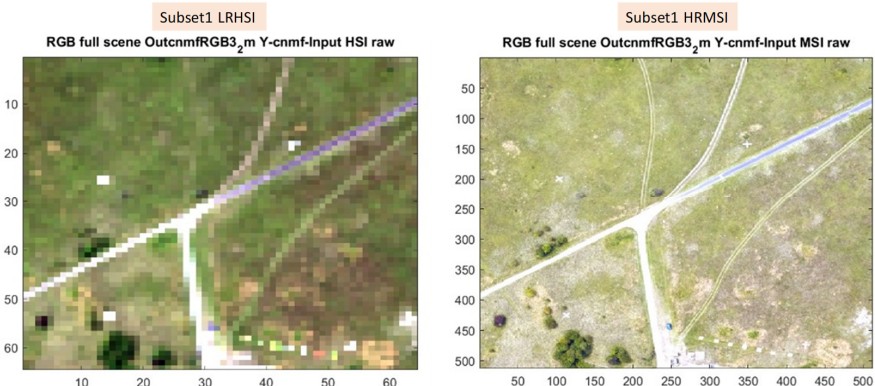

**Figure 9.** The RGB representation of the LRHSI (3.2 m GSD) of subset 1 (**left**) and its corresponding HRMSI at 0.5 m GSD (**right**) for the input of the sharpening algorithms. Note that all manmade targets, even the large calibration panels at the bottom of the scene, were fuzzy geometrically in the LRHSI input data.

3.3.1. Subset 1: Performance of SR Assessed by L1NE

Figure 10 presents the L1NE assessments of SR outputs produced by the proposed SR algorithms and compares the SR performance with the results produced by Yokoya's original algorithm (Y-CNMF) through the fused coupled decomposition of the 3 m GSD LRHSI subset 1 and the RGB-HRMSI datasets. It is seen from the figure that the three proposed SR algorithms exhibited ~2 times smaller average L1NE compared with Y-CNMF. Figure 11 presents the mean spectral plot of various ROIs of the sharpened scenes produced by all algorithms in this work. Similar to the subset 0 data, it can be seen that Yokoya's Y-CNMF produced worse spectral accuracy compared with the proposed SR algorithms. The better accuracy of the sharpened scenes in the proposed algorithm is ascribed to the various R-SRC estimation enhancements, which are absent in the Y-CNMF method. The results in Figures 4 and 10 thus consistently demonstrate the significance of the R-SRC estimation for enhancing the output of the fusion-based SR process.

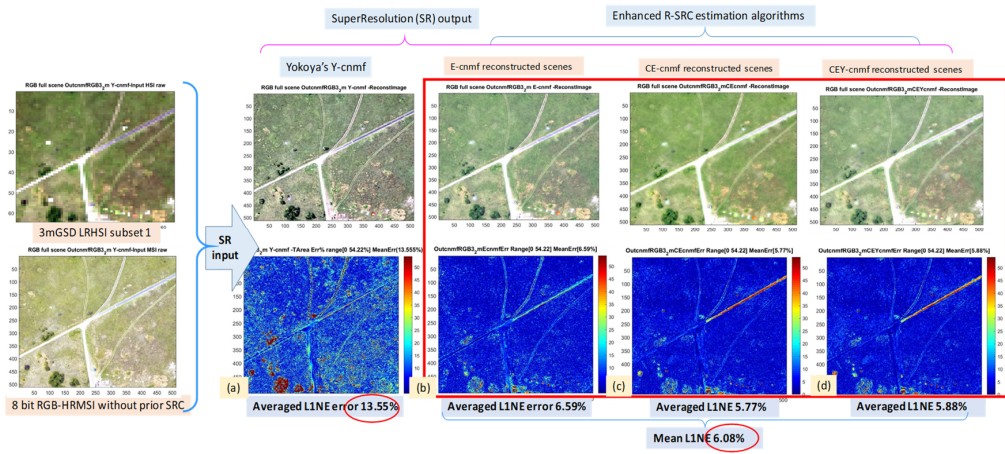

**Figure 10.** The performances of SR through the fusion of the 3 m GSD LRHSI of subset 1 with the RGB-HRMSI using all SR algorithms proposed in this work. First column: RGB representation of the input datasets. Upper panels (**a**–**d**): the RGB representations of the outputs of the four SR algorithms: Yokoya's original algorithm Y-CNMF, the proposed 3 algorithms in red box: E-CNMF, CE-CNMF, and CEY-CNMF, respectively. Lower panel: the corresponding L1NE w.r.t. the GT for the SR outputs obtained from the four SR algorithms, in which the proposed algorithms (**b**–**d**) exhibited a factor of ~2-fold error reduction compared with Yokoya's original algorithm, shown in (**a**).

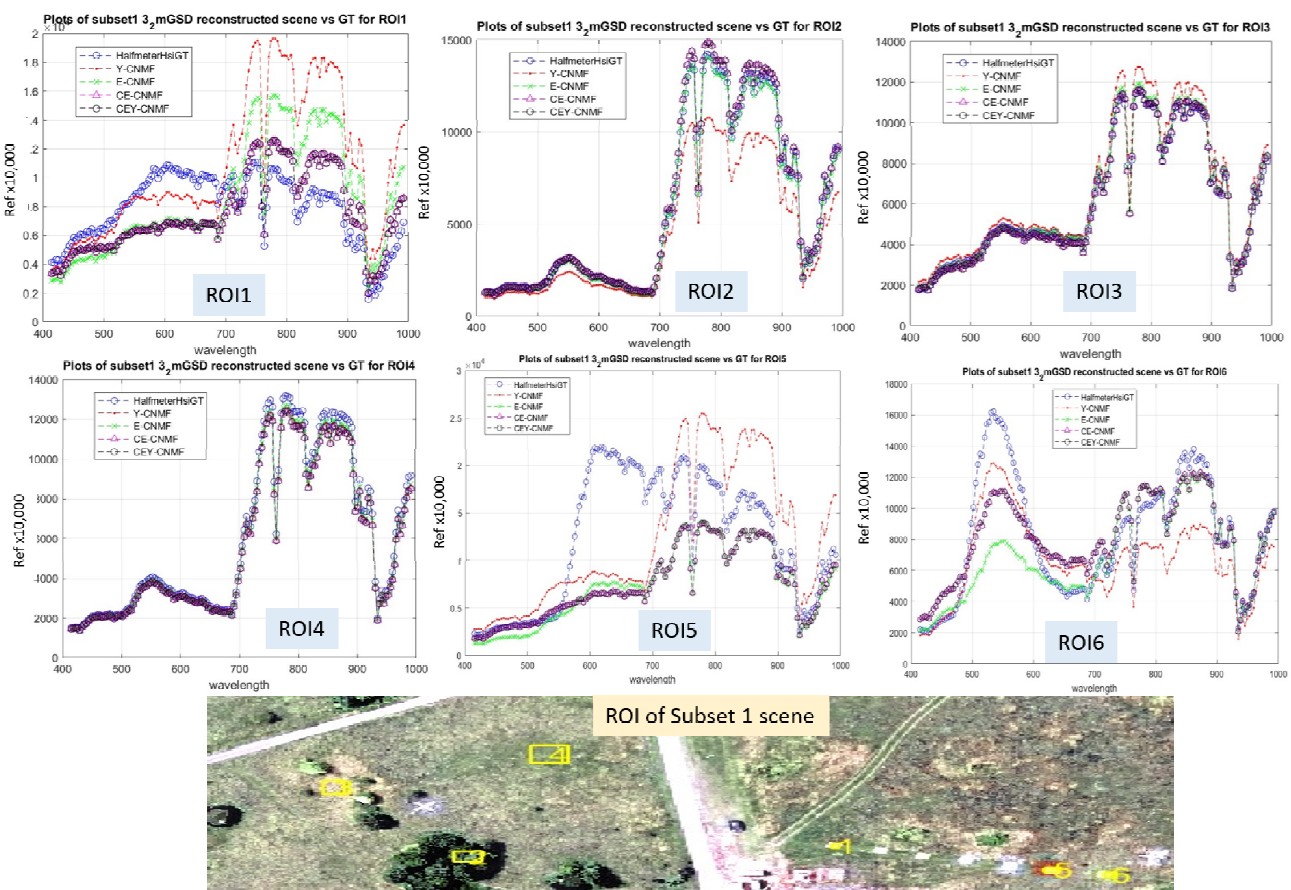

**Figure 11.** The spectral characteristics of pixels for a selection of natural vegetation and manmade materials reconstructed by all proposed sharpening algorithms for the subset 1 3 m GSD LRHSI, compared with results from Yokoya's original algorithm (plotted in red dots) and the ground truth (GT; plotted in blue circles). It is seen that the proposed algorithms, particularly the CEY-CNMF, performed much better than Yokoya's algorithm, with better spectral resemblance to the GT.

3.3.2. Subset 1: ROC Assessment for the Recovery of Large Targets with Abundance 0.44

Similar to Section 3.2.2, this section examines the target detection performance for the larger targets in the sharpened scenes of subset 1. There are large 2 × 2 m calibration panels as well as small ceramic tiles of 0.2 × 0.2 m in size scattered around in subset 1. The ROC behavior was interpreted from the ACE detection statistics with respect to the target maps shown in Figure 2, which contain both large and small targets in this dataset. Figure 12 depicts the ROC for the ACE detection of seven manmade calibration panels from sharpened images produced by all SR algorithms used in this work. It is seen that the ROC for some panels, such as the orange and green Perspex (ROI 6 and 3), achieved a several orders of magnitude better false alarm rate, for certain detection rates, for the proposed enhanced R-SRC algorithms, compared with Yokoya's algorithm. The exact reason why the enhancement of detection was not the same across all different types of targets is not known at present. Figure 13 shows the averaged ROC for all seven targets in subset 1, and it is clear that the proposed two-level CEY-CNMF provided an order of magnitude false alarm reduction in the low probability of false alarm region, compared with Yokoya's algorithm. The better detection of targets from the scene indicates a more accurate and faithful reconstruction of the HRHSI.

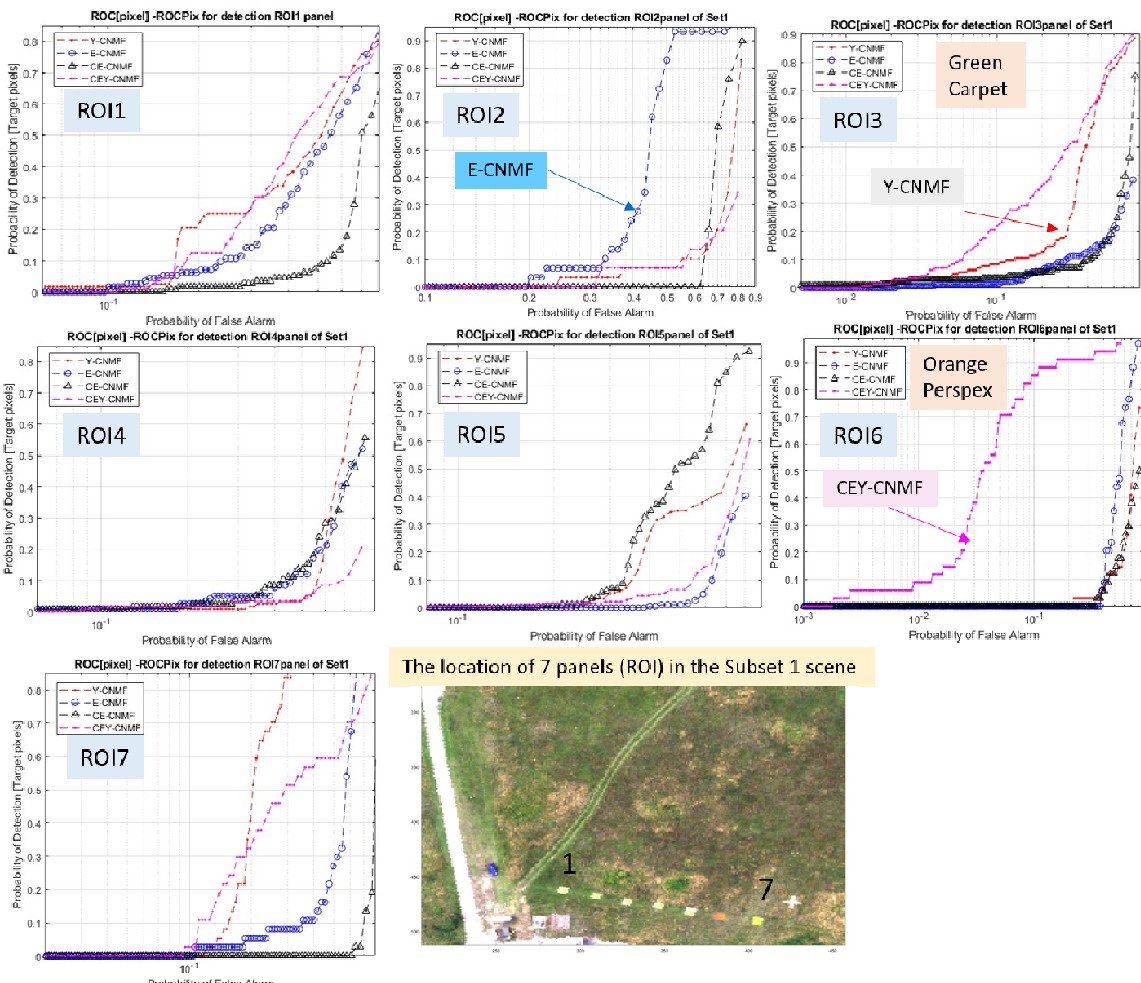

**Figure 12.** The ROC for the ACE detection of seven manmade calibration panels from the sharpened images processed by the proposed algorithms and compared with results from Yokoya's original algorithm (in the red dot plot). It is seen that the ROC of some panels, such as the orange Perspex (ROI6), exhibited a several orders of magnitude better false alarm rate, at detection rates of about 0.7, for the proposed algorithms compared with Yokoya's original algorithm. The RGB picture depicts the location of the ROI 1-7.

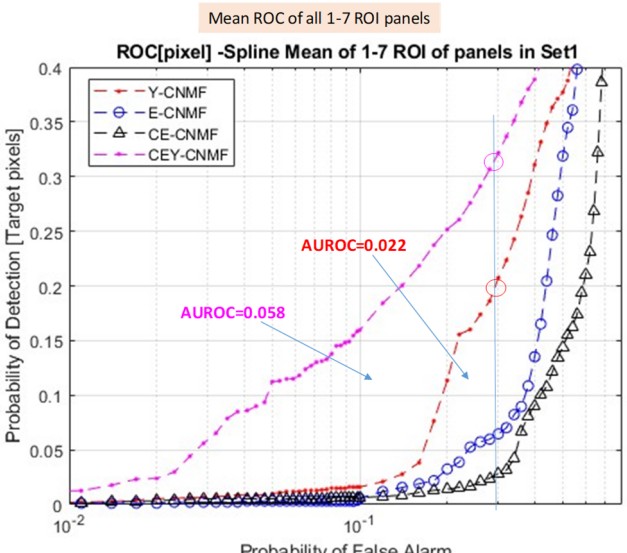

**Figure 13.** The mean of the ROC for the ACE detection over seven manmade calibration panels as targets. It is seen that the proposed two-level SR algorithm, CEY-CNMF (in the magenta plot), provided an order of magnitude false alarm reduction in the low PFA region, with an over 100% better area under the ROC (AUROC) between a PFA of 0.01 and 0.3, compared to Yokoya's algorithm (in the red dot plot).

### 3.3.3. Subset 1: ROC Assessment for the Recovery of Small Targets with Abundance of 0.015

This section examines whether single-pixel-size targets in the 0.5 m GSD HRHSI could be recovered from the aggregated 3.2 m GSD LRHSI scene through the fusion-based SR approach. The aggregated GSD 3.2 m LRHSI of subset 1 contained small green ceramic targets of sizes $0.2 \times 0.2$ m, which were mixed with the background with effective abundances of about 0.015. The RGB representation of the GSD 3.2 m LRHSI, as shown in Figure 9, appeared to be fuzzy throughout the entire scene, including large structures, such as the road path.

Figure 14 depicts the ROC for the detection of green ceramic tiles as targets from the aggregated subset 1, and it shows that the scene that was reconstructed by the proposed CEY-CNMF (in the magenta plot) was superior to that sharpened by Yokoya's original algorithm (in the red dot plot). The better detection of targets from the scene implies a more accurate reconstruction of the HRHSI. However, the ROC that is displayed in Figure 14 could be strongly skewed by the presence of 'easy' targets in the scene, such as the presence of the large calibration panel, which had the same spectral signature as the small green ceramic tiles. To ascertain whether the small green ceramic targets, which have abundances of 0.015 in the LRHSI, could be recovered after the SR sharpening, the detection rate at a certain arbitrarily chosen probability of false alarm (PFA), specifically 0.1 PFA, was assessed. The false color map of the detections for the scene sharpened by the proposed CEY-CNMF and Yokoya's Y-CNMF algorithms at the PFA of 0.1 are shown in the lower panel of Figures 14b and 14c, respectively. The color code of the false color detection map is as follows: correctly detected small green ceramic tile target pixels are labeled in white and are also circled in red for easy visualization, missed targets are in magenta, and false alarms are in green. It is seen from the figure that the scene that was sharpened by the proposed CEY-CNMF algorithm managed to recover 14 green ceramic targets, while Yokoya's algorithm could only find 1.

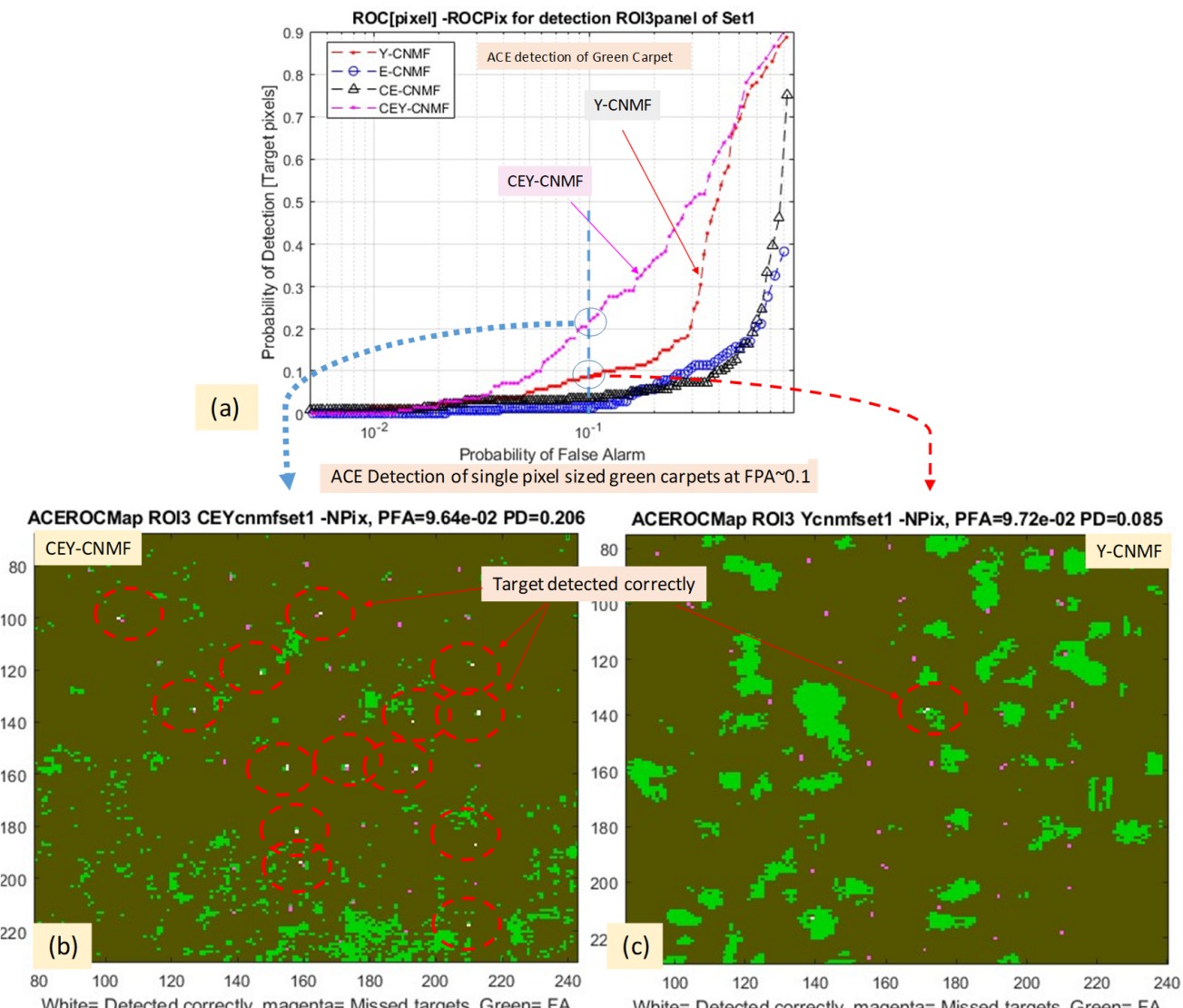

**Figure 14.** The effectiveness of the proposed R-SRF enhanced CNMF algorithm for the recovery of small occupancy (~0.015) green ceramic targets from the 3.2 m GSD LRHSI scene. (**a**) The ROC for the detection of the green ceramic targets by all algorithms. Lower panel: the false color detection maps at a PFA of 0.1 for the scenes sharpened by CEY-CNMF (in (**b**)), which exhibited over an order of magnitude better target detection performance than Y-CNMF (in (**c**)).

From Figure 14, the question arises as to why the proposed enhanced R-SRC algorithm was capable of recovering highly mixed targets of abundance ~0.015. Furthermore, it is necessary to understand how the proposed CEY-CNMF algorithm managed to provide an improvement in performance of over an order of magnitude compared to Yokoya's Y-CNMF algorithm for the recovery of these small sub-pixel targets. To this end, Figure 15 depicts the spectral characteristic plot of pixels at randomly selected locations of the green ceramic tiles in the subset 1 scene, which was sharpened by all algorithms. It is seen that the spectra of the target pixels for the scene sharpened by Y-CNMF exhibited relatively larger spectral discrepancy w.r.t. that of the GT, presumably due to the insufficient R-SRC estimation in Yokoya's original algorithm.

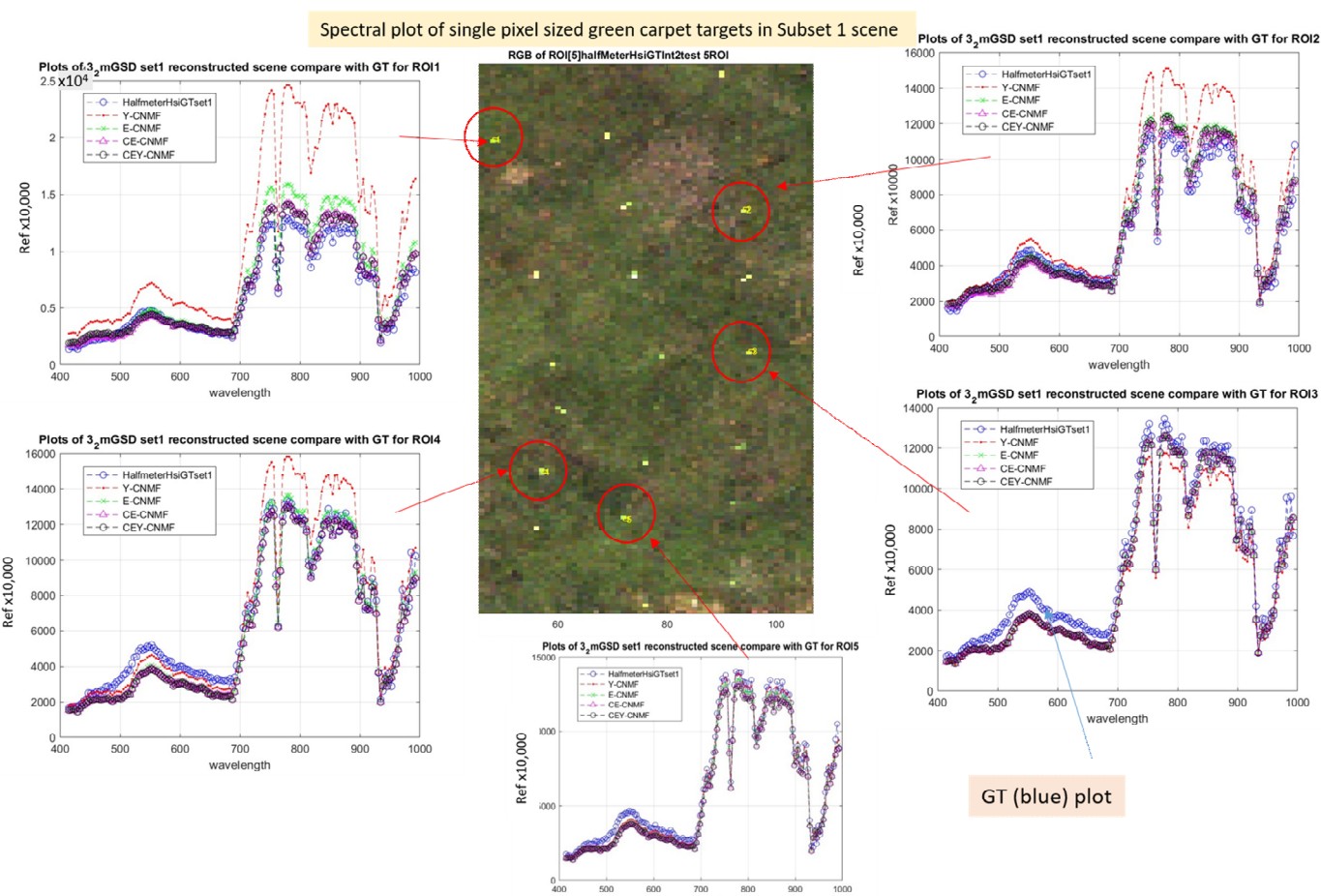

**Figure 15.** Spectral plots of randomly selected pixels at the location of small green ceramic targets for the subset 1 scenes sharpened by all SR algorithms are shown. It is seen that the sharpened scene processed by Yokoya's original algorithm (in the red dot plot) exhibited relatively larger spectral discrepancy w.r.t. the GT than those sharpened by the proposed enhanced R-SRC algorithms.

## 4. Discussion

The main objective of the present work was to examine whether a low-cost HSI system together with a mega-pixel RGB HRMSI camera will be effective for long-range surveillance applications through the use of SR technologies. The theme requires the understanding of the current state-of-the-art machine learning capabilities, to identify the technology gap and to propose the way forward for the realization of the set objective. In this respect, a real HSI scene of 0.5 m GSD spatial resolution was used as the HRHSI ground truth (GT), which was then resampled into 3 m GSD LRHSI for the present experiment. A mega-pixel three-band RGB dataset of the scene with unknown imaging hardware information was employed as the HRMSI for sharpening the LRHSI. Three variants of a modified CNMF scheme based on Yokoya's Y-CNMF algorithms [17–19] were utilized for the experiment. The modification of the algorithm in this work consisted exclusively of the enhancement of R-SRC estimation between the LRHSI and HRMSI datasets, which was designed to enhance the spectral matching between the input dataset pairs for improving the quality of sharpening. Yokoya's Y-CNMF algorithm is considered as the baseline algorithm, and it does not provide sufficient R-SRC estimation capability within the algorithm.

The paper first outlined the significance of R-SRC to the fusion-based SR processing through a simple experiment using two different spectral scales of the same HRMSI datasets for sharpening a 3.2 m GSD LRHSI scene. These two sets of data were then sharpened by Yokoya's Y-CNMF algorithm. The reconstruction errors (L1NE) of these two sharpened scenes were found to be substantially different, with one of them almost double the other,

yielding a ~24% error when the RGB HRMSI was used as the input for the SR (Figure 3). This experiment vividly outlined the significance of spectral calibration between the LRHSI and HRMSI, and showed that it can be achieved through the R-SRC, such that the spectral characteristics of LRHSI and HRMSI can be closely matched before they undergo coupled decomposition for SR. This result thus hints at the real need for enhancing the estimation of R-SRC in the fusion-based SR process. In this respect, three variants of R-SRC enhancements based on Yokoya's core CNMF algorithm were proposed in this paper.

Instead of using SAM for assessing the spectral integrity of the sharpened scene, the better assessment metric of L1NE, which is more suitable if the ultimate application is target detection, was demonstrated in Figures 4 and 5. SAM yields the spectral angle difference between two vectors, which is not sufficient information to predict utility for target detection purposes. Target detection methods, especially those that utilize prior information about the target, such as that in the ACE detector, require the magnitude of the vector for detection, too. Thus, spectral norm differences of pixel vectors are critically important for target detection. SAM exhibited results that were only ~20% better for the proposed algorithms than the original Y-CNMF, while the L1NE assessment shown in Figure 4 indicated a ~3-fold reduction of spectral errors for the SR reconstructions by the proposed algorithms. Thus, instead of using SAM for assessing the quality of the SR results, the effectiveness of the R-SRC for enhancing the quality of sharpened scenes was exclusively validated through the L1-norm-error (L1NE) and the receiver operating characteristics (ROC) in this work. Considering the quality of sharpening of two subsets of the real vegetation scenes (Selene) by using all four algorithms, as revealed by the L1NE (in Figures 4 and 10) and ROC (Figure 7, Figure 12, and Figure 14) assessments, it was seen that the results of the scene sharpened by the proposed R-SRC enhanced algorithms were superior, with 2- to 3-fold reductions in errors compared with the baseline Y-CNMF algorithm. Experiments also revealed that the sharpened scenes processed by the proposed R-SRC enhanced algorithms were capable of recovering very small targets of 0.015 occupancy in the LRHSI data (Figure 14), with an order of magnitude higher detection rate, for a particular false alarm rate, than the baseline Y-CNMF algorithm.

The encouraging results obtained from this work suggest that long-range surveillance using low-cost HSI systems could be realized in the near future, providing the following issues can be understood and subsequently addressed:

- It appears that some objects in the scene, such as those 'rare' species, such as the man-made foot path and panels, exhibited larger reconstruction errors than the vegetation species (see Figures 4, 6 and 10). This might be due to the relatively small number of spectra of these materials in the multidimensional simplex enclosing the image spectra, which induced local mini-max when the $\mathbf{E}$ or $\mathbf{E_m}$ in Equation (15) were evaluated. It is possible that additional processing, such as the partition of the scene through SLIC clustering [41], will be able to help solve this problem.
- It is intriguing to note the rather variable ROC statistics over the seven manmade panels that were recovered by the proposed algorithms: some panels appeared to be recovered much better than others, as revealed by the ROC (Figure 12). It is worthwhile to look deeper into whether this is caused by some specific spectral characteristics of the panels, or whether it is due to other factors that affect the accuracy of the reconstruction of the panels' spectral properties.
- It is noted that the false alarm rate for target detection with sharpened data (Figure 14) was much higher (~one or two orders of magnitude) than for detection using the GT HRHSI (for a comparable detection rate). This may suggest that the CNMF algorithm and the R-SRC estimation both need to be improved in order to realize this technique for real-world applications [6]. However, it should also be noted that the targets exhibited an extremely low abundance in the LRHSI, making it unsurprising that detection was difficult.

## 5. Conclusions

This paper concerns the effectiveness of super-resolution (SR) techniques for spatial sharpening of low-spatial-resolution hyperspectral imaging (LRHSI) data into high-spatial-resolution HSI (HRHSI). The fusion-based SR approach, which utilized coupled non-negative factorization (CNMF) of LRHSI together with geographically registered high-spatial-resolution multispectral imaging (HRMSI), was arbitrarily chosen for this work. The HRMSI data that were employed here were acquired by another imaging system, and its optical transfer characteristics are not known. This means that the color scheme (i.e., spectral characteristics, such as bandwidth and spectral response) of the HRMSI could be very different from that of the narrow spectral bands of the LRHSI. Thus, the coupled decomposition of both datasets, without performing appropriate pre-adjustments for matching the spectral characteristics of both, is expected to induce large spectral errors in the final sharpened output. To address this issue, three variants of enhanced relative spectral response characteristics (R-SRC) algorithms were derived for matching the spectral characteristics of the LRHSI and HRMSI before they were fused by CNMF. The modified algorithm was based on the core of Yokoya's Y-CNMF algorithm, which is considered as the baseline algorithm to compare with.

Through L1-norm-error (L1NE) and receiver operating characteristic (ROC) assessments, experiments showed 2- to 3-fold error reductions in the scenes that were sharpened by the proposed R-SRC algorithms, in comparison to the baseline Y-CNMF algorithm. Experiments also revealed that the sharpened scenes that were processed by the proposed R-SRC enhanced algorithms were capable of recovering very small targets of 0.015 occupancy from the LRHSI data, with an order of magnitude higher detection rate, for a particular false alarm rate, than the baseline Y-CNMF. The seemingly impressive results presented in this paper may signify the importance of R-SRF, particularly when a fusion-based approach is utilized for SR. However, the present work by no means suggests that research on R-SRC alone is sufficient for SR technology to be deployed for real applications. There are a number of observations from the data, such as the substantial reconstruction errors and the high false alarm rate from 'rare' species in the scene, that need to be addressed as the next stage of research. Due to the page limitation of the present paper, R-SCR estimations using other approaches, such as Lanaras's approach [28], will be considered in our next stage of research.

**Author Contributions:** Conceptualization, J.P., C.Y., and P.Y.; methodology, P.Y.; software, C.Y. and P.Y.; validation, J.P. and C.Y.; formal analysis, P.Y.; investigation, J.P. and C.Y.; resources, J.P.; data curation, M.C.; writing—original draft preparation, P.Y.; writing—review and editing, C.Y. and J.P.; visualization, C.Y.; supervision, P.Y.; project administration, P.Y.; funding acquisition, J.P. All authors have read and agreed to the published version of the manuscript.

**Funding:** This research has not received research funding.

**Data Availability Statement:** Data are contained within the article.

**Acknowledgments:** The author J.P. acknowledges the studentship support by DSTL, C.Y. acknowledges the Counter-Terrorism Centre of DSTL for providing time to conduct work on this project and M.C. acknowledge the Turkey government for providing the PhD studentship to work with Cranfield University.

**Conflicts of Interest:** The authors declare no conflicts of interest.

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
