# Peer review of "Enhanced Hyperspectral Sharpening through Improved Relative Spectral Response Characteristic (R-SRC) Estimation for Long-Range Surveillance Applications"

_electronics, doi:10.3390/electronics13112113_

Round 1

Reviewer 1 Report

Comments and Suggestions for Authors

1. On page four, is Y_{h_i} in Equation 6 obtained using the spatial point spread function S? Does the proposed method require prior knowledge of the point spread function when applied?

2. The description from lines 186 to 190 on page four and Equation 7 is confusing. Is R the spectral response function? If so, the data dimensions of R and Y in Equation 7 are different. How can they be subtracted?

3.The "iff" in Equation 7 should be "if", and y_i needs to be defined.

4. The formatting (font, paragraphs) in the paper is inconsistent. For example, the font and margins in lines 180-184 on page four differ from the preceding text.

5. The paper proposes an improved method to estimate the spectral response function. To validate the effectiveness of the proposed method, the experimental results should demonstrate the discrepancy between the estimated spectral response functions and the ground truth.

6. The resolution of the images in the experimental results is low, which does not provide clear visualization.

7. The experimental results in Section 3.1 could not demonstrate the importance of spectral response functions for fusion well. After normalization with Equation 17, the proportional relationship among the three bands of each pixel in the RGB image remains unchanged, while the grayscale values of different pixels change. Therefore, Equation 17 introduces changes in the spatial dimension rather than solely altering the estimation of spectral response functions.

8. Add quantitative metrics to the experimental results, such as the average values of SAM and L1NE and the area under the ROC curves.

9. The proposed method requires LR-HSI and HR-MSI to be co-registered in the spatial domain. How can fully co-registered LR-HSI and HR-MSI be obtained in practical applications?

Reviewer 2 Report

Comments and Suggestions for Authors

This paper proposes an enhanced algorithm for estimating the Relative Spectral Response Characteristics (R-SRC) between Low Spatial Resolution Hyperspectral Images (LRHSI) and High Spatial Resolution Multispectral Images (HRMSI) for super-resolution (SR) applications. By improving R-SRC estimation, the study introduces three variant algorithms to enhance the accuracy of the image fusion process, thereby more effectively recovering HRHSI. I recommend publishing the article after minor revision.

1.     Detailed Description of Spectral Response Characteristics: Although the improved R-SRC algorithm is mentioned, the description of how spectral feature matching between LRHSI and HRMSI is achieved, as well as the physical or technical basis for this matching, is not sufficiently detailed. I suggest adding more technical details, such as specific methods and steps of spectral matching.

2.     Depth of Algorithm Validation: The paper mentions using experimental data to validate the effectiveness of the algorithm, but information about data selection, experimental settings, and control variables is scarce. I recommend supplementing with detailed information on these experimental designs, including specific conditions, selection criteria for the datasets used, and control variables during the experiment, to enhance the reliability and generalizability of the results.

3.     Breadth of Comparative Analysis: The paper only compares with Yokoya's Y-CNMF algorithm. To strengthen the persuasiveness of the research, I suggest comparing several other latest or widely used related technologies to demonstrate the relative position and advantages of your method in the current technological context.

Author Response

Journal  Electronics (ISSN 2079-9292)

Manuscript ID electronics-2973865

Type Article

Title: Enhanced Hyperspectral sharpening through improved relative spectral response characteristic (R-SRC) estimation for long range surveillance applications

Authors: Jonathan Piper , Catherine Yuen , Mehmet Fatih Cakir , Peter Yuen * 

The authors would like to thank the reviewer’s constructive comments for improving the clarity of the paper. The point to point response to the reviewer’s comment is highlight in red in this document, and the corresponding amendments in the paper is depicted in red font with tracked changes in the file ‘Enhanced Hyperspectral sharpening PY R1 30-4-24.docx’. 

The reviewer’s comment is presented in black font.

Review Report (Reviewer 2)

Comments and Suggestions for Authors

This paper proposes an enhanced algorithm for estimating the Relative Spectral Response Characteristics (R-SRC) between Low Spatial Resolution Hyperspectral Images (LRHSI) and High Spatial Resolution Multispectral Images (HRMSI) for super-resolution (SR) applications. By improving R-SRC estimation, the study introduces three variant algorithms to enhance the accuracy of the image fusion process, thereby more effectively recovering HRHSI. I recommend publishing the article after minor revision.

Response: Thank you for the comment and we’d like to address all the comments raised by the reviewer in the following paragraphs.

Respond to Reviewer 2:

  1. Detailed Description of Spectral Response Characteristics: Although the improved R-SRC algorithm is mentioned, the description of how spectral feature matching between LRHSI and HRMSI is achieved, as well as the physical or technical basis for this matching, is not sufficiently detailed. I suggest adding more technical details, such as specific methods and steps of spectral matching.

Response: Thank you for the excellent question and in fact there are two levels of the spectral matching between the LRHSI and the HRMSI in this work. The first level is the spectral matching between LRHSI and the reduced spatial resolution of HRMSI (denoted as Yh in Eq 5), and this process is commonly termed as ‘relative spectral response character (R-SRC) matching and it has been the main theme that has been focused in the present paper. This part of spectral matching was achieved through quadratic programming via Eq 7 (Yokoya’s base line algorithm), and also Eqs 8,9, and 10  which were the 3 proposed algorithms in the present paper.

The second level of spectral matching was achieved through the decomposition of the LRHSI and HRMSI jointly in 2 nested routines as described in Eqs 11-15, for the extraction of the endmember E and the abundance A, from the spectrally degraded endmember Em and the spatial degraded abundance Ah. Please refer to L288-313 for more details of this part of algorithm. Note that this second part of spectral matching is the same as Yokoya’s base line algorithm, and only the first level of spectral matching is different.

To clarify the point raised by the reviewer,  the following sentences have been added in L166 of the paper:

The objective of Yh is the formation of a spatially degraded resolution Y such that it is pixel matched with the LRHSI, to allow the spectral responses R between these two sets of data to be deduced over the entire scene. Note that this is the first level of spectral matching between the LRHSI and the HRMSI.

Also title heading of section 2.3 (L248) is modified to:

2.3 Implementation of CNMF for spectral unmixing: 2nd level of spectral matching between LRHSI and HRMSI

  1. Depth of Algorithm Validation: The paper mentions using experimental data to validate the effectiveness of the algorithm, but information about data selection, experimental settings, and control variables is scarce. I recommend supplementing with detailed information on these experimental designs, including specific conditions, selection criteria for the datasets used, and control variables during the experiment, to enhance the reliability and generalizability of the results.

Response: Thank you for the question and in fact the data set that had been utilized in this work was collected by the UK DSTL as part of their airborne countermeasure project. The details of the experimental settings, data selection and control variables had been documented in ref 37 (Piper, J. ‘A new dataset for analysis of hyperspectral target detection performance’ , Hyperspectral Imaging and Applications Conference, October 2014,Coventry UK., 2014). The general conditions for experimental designs had been documented in details in ref 4 (Yuen PWT, Richardson M, ‘An introduction to hyperspectral imaging and its application for security, surveillance and tar-get acquisition’, The Imaging Science Journal 58 (5), 241-253, 2010). Interested readers can refer to these documents for more information with respected to the data sets that had been used in this work.

To clarify this point the following sentences have been added in L 334:

Please refer to references  [4,37] for more information about the properties of the presently utilized data sets such as the experimental settings, design of experiment, control variables and data selections.

  1. Breadth of Comparative Analysis: The paper only compares with Yokoya's Y-CNMF algorithm. To strengthen the persuasiveness of the research, I suggest comparing several other latest or widely used related technologies to demonstrate the relative position and advantages of your method in the current technological context.

Response: Thank you for raising this question and in fact the study of R-SRC in fusion based super-resolution (SR) research is severely lacking. Most if not all, of the work done in the past was performed through simulation studies where the HRMSI was extracted from the ground truth (GT) of the HSI data set. In these cases the spectral matching between the LRHSI and the HRMSI would be perfectly excellent! However this is not the case in practise. The present work was part of the DSTL programme for exploring whether SR may be useful in practise for the long range surveillance application, without the need of expensive optics in the airborne/spaceborne instruments. Yokoya’s work adopted in here had been served as the base line algorithm with the sole purpose to highlight the significance of the R-SRC to the accuracy of the spatially sharpened HSI data when they were processed under the fusion based SR algorithm. The present results revealed vividly that the estimation of R-SRC could enhance the goodness of the spatially sharpened LRHSI 3 times better than that of the base line algorithm. The other work in this discipline had been Lanaras’s work (ref 28) which utilized prior information of the HRMSI hardware, which was hardly available in practise. Due to the lack of related work in the field and also the page length limitation of the paper, other work will be published in our forthcoming paper.

To clarify this point the following sentences have been added in L709:

Due to the page limitation of the present paper, R-SCR estimations using other approaches such as Lanaras’s [28], will be considered in our next stage of research.  

Reviewer 3 Report

Comments and Suggestions for Authors

This research paper focuses on the fusion of low spatial resolution hyperspectral images (LRHSI) and high spatial resolution multispectral images (HRMSI), a topic that has wide-ranging applications and has long attracted the attention of researchers in the field. The research team addresses the need for matching the spectral characteristics before decomposition in LRHSI and HRMSI. The three new algorithms for Relative Spectral Response Characterization (R-SRC) proposed by the research group pave the way for potential solutions to this problem.

1.In the Introduction, the study of super-resolution through both hardware and software methods is mentioned. There's a need to elaborate on how these two aspects enhance the spatial resolution of HSI, as well as their respective advantages and disadvantages.

2.The framework of the Introduction needs modification. The last paragraph should be descriptive rather than merely listing each chapter. It's recommended to incorporate the two major research objectives into the last paragraph to provide clarity for the reader.

3.All figures in the article need brief titles for summarization, and then further explanations within the text. In particular, Figure 15 lacks clarity and there's an issue with the coordinates being obscured in the top-left corner.

4.Please check the format of all references, ensuring that they adhere to the citation format required by this journal.

Comments on the Quality of English Language

In order to make the manuscript more suitable for journal publication, the English aspect needs to be modified and improved to minimize the use of long sentences and reduce grammatical errors.

Reviewer 4 Report

Comments and Suggestions for Authors

In this manuscript, the authors proposed a method to improve the effectiveness of super-resolution (SR) technique for spatial 672 sharpening of low spatial resolution hyperspectral imaging (LRHSI) data into high spa-673 tial resolution HSI (HRHSI). The results of the experiment indicate that the method has some effectiveness, but there are structural and language issues in the writing of the article. I suggest making some revisions before it can be accepted for publication. Here I list some of the points.

1.     The fonts and formatting throughout the entire text are inconsistent, for instance, the section from line 176 to line 210.

2.     There are many formulas in the Section 2.2 and 2.4, that’s good for the understanding of the data processing. However, the derivation of the formulas identical to those in the reference can be omitted.

3.     The figure titles are too long and cannot clearly be distinguished from the main text.

4.     Both the titles and the main text are often ended with “!”, that’s quite weird.

5.     The upper frame of Figure 6 is not displayed complete.

6.     The "Discussion" section contains lengthy descriptions of previous results and the content of the article, which need to be reorganized. This part written by the author resembles more of a prospect rather than a discussion analysis.

Comments on the Quality of English Language

As suggested, it would be preferable to have a native speaker review and polish the language and logical structure of the article.

Round 2

Reviewer 1 Report

Comments and Suggestions for Authors

The resolution of the images in the paper is low. For example, the text in Figures 11 and 12 is not clear. Please use images with higher dpi. 

Reviewer 3 Report

Comments and Suggestions for Authors

All images must be presented in a clearer form and graphics must see the coordinate system, especially Figures 1, 3, 4, 6, 11.

Please do not include references in the abstract.
